

# Classicality with(out) decoherence: Concepts, relation to Markovianity, and a random matrix theory approach

**Philipp Strasberg**[⋆]

Física Teòrica: Informació i Fenòmens Quàntics, Departament de Física,
Universitat Autònoma de Barcelona, 08193 Bellaterra (Barcelona), Spain

⋆ philipp.strasberg@uab.cat

## Abstract

Answers to the question how a classical world emerges from underlying quantum physics are revisited, connected and extended as follows. First, three distinct concepts are compared: decoherence in open quantum systems, consistent/decoherent histories and Kolmogorov consistency. Second, the crucial role of quantum Markovianity (defined rigorously) to connect these concepts is established. Third, using a random matrix theory model, quantum effects are shown to be exponentially suppressed in the measurement statistics of slow and coarse observables *despite* the presence of large amount of coherences. This is also numerically exemplified, and it highlights the potential and importance of non-integrability and chaos for the emergence of classicality.



# 1 Introduction

It is an obvious everyday fact that the world around us does not show direct quantum effects: we can safely disregard the wave-like behaviour of matter and do not need to worry about the effects of measurement backaction. But this causes a conundrum because our everyday world is built out of particles that are fundamentally quantum. Studying the emergence of classicality from underlying quantum physics is thus of foundational importance, but also has great practical relevance for the realization of quantum technologies. Yet, the questions *how* to prove the emergence of classicality and also the prior question *what* needs to be proved are not fully settled. The present paper aims to clarify and extend the discussion and it is divided into two parts.

The first part (Sec. 2) is about clarifications and makes two contributions. The first contribution is to give a focused overview over three different approaches to the question *what* needs to be proved. These are the decoherence approach in open quantum systems (OQS) [1–4], the consistent/decoherent histories formalism [5,6], and a recent approach based on the notion of Kolmogorov consistency [7–15]. The second contribution emphasizes the crucial role played by the rigorous definition of (multi-time) quantum Markovianity [16–18] (for introductions see Refs. [13, 14]), which connects the decoherence approach in OQS to the other two approaches.

The second part (Sec. 3) is about *how* to prove classicality by extending recent research results [7, 8, 15]. Therein it has been observed that isolated many-body systems can behave classical even in presence of large amount of coherences and that non-integrability and chaos might be the key to understand this behaviour. In particular, Ref. [15] argued that this is a generic effect for a large class of observables (specified in greater detail later on) and estimated that deviations from classical behaviour are exponentially small in the system size. Here, we confirm this estimate and provide an alternative derivation of it in Sec. 3.1, which is inspired by the idea to model complex isolated quantum systems by random matrix theory [19–24]. Moreover, a simple model is used to illustrate important features of this new approach to classicality in Sec. 3.2, before concluding in Sec. 4.

For completeness, we remark that there are alternative explanations of classicality, which we will not study here. For instance, classicality is sometimes explained by gravity as the fundamental cause for decoherence [25] or by collapse theories that directly modify Schrödinger's equation [26]. However, we are here interested in explanations *within* the conventional framework of non-relativistic quantum mechanics, where gravity or collapse theories cannot play any role. Moreover, also the semiclassical limit of large action $S/\hbar \gg 1$—formalized by taking $\hbar \to 0$ [27] among other limiting procedures related to temperature, mass, angular momentum, etc.—is often invoked to explain classicality. While this provides an important consistency check, we here avoid such limiting procedures (after all, decoherence is a major obstacle to build a quantum computer and one can certainly not claim that a quantum computer operates in the high temperature limit). Finally, for fundamental criticism about decoherence we refer the reader to Leggett [28], the importance of decoherence for macroscopic objects was discussed in Refs. [29–34], and for a general criticism of prevailing notions of classicality in the cosmological context see Ref. [35]. Furthermore, a topic which we will only briefly touch is quantum Darwinism [36–38], which further refines the notion of decoherence in OQS.

## 2 Classicality: Definitions and approaches

### 2.1 Definition used in this work

For any discussion of the quantum-to-classical transition, it is important to precisely define what "classical" behaviour means. One here hits the first obstacle as the boundary between the quantum and classical world is *not* one-dimensional: depending on the problem different boundaries can be drawn.

For instance, one legitimate way to define classicality could be to ask whether a state of a bipartite quantum system obeys all Bell inequalities or not. This definition, however, only reveals *static* quantum features of a *state*, and does not allow to draw any conclusion about how classicality *emerges* from an underlying quantum description.

Here, we are precisely interested in this emergence and ask whether a *process*, i.e., an experimentally well-defined procedure to access the time evolution of a quantum system [13, 14], can look classical. To be precise, consider an isolated quantum system with (time independent) Hamiltonian $H$ prepared in some state (density matrix) $\rho_0$. Let $X = \sum_{x=1}^{M} \lambda_x \Pi_x$ be some observable with eigenvalues $\lambda_x$, eigenprojectors $\Pi_x$ and $M$ denoting the total number of distinct eigenvalues (measurement outcomes). The probability to measure $x_n, \ldots, x_1$ at times $t_n > \cdots > t_1$ is

$$p(x_n, \ldots, x_1) \equiv \text{tr}\left\{\Pi_{x_n} U_n \ldots \Pi_{x_1} U_1 \rho_0 U_1^{\dagger} \Pi_{x_1} \ldots U_n^{\dagger}\right\}, \tag{1}$$

with $U_k \equiv e^{-iH(t_k - t_{k-1})}$ the unitary time evolution operator between two times ($\hbar \equiv 1$). Note that Eq. (1) can be experimentally reconstructed by performing $n$ repeated measurements on a quantum system and by repeating this procedure many times to create sufficient statistics. Next, pick some $k \in \{1, \ldots, n-1\}$ and define $p(x_n, \ldots, \cancel{x_k}, \ldots, x_1)$ to be the same probability as in Eq. (1) *except* that no measurement is performed at time $t_k$ (and thus no outcome $x_k$ is recorded), which is indicated with the notation $\cancel{x_k}$ and obtained from Eq. (1) by dropping the two projectors $\Pi_{x_k}$. Then, the process is classical if the following "probability sum rule" is satisfied for all $k < n$ and all $n > 1$ (up to some error much smaller than the considered probabilities):

$$\sum_{x_k} p(x_n, \ldots, x_k, \ldots, x_1) = p(x_n, \ldots, \cancel{x_k}, \ldots, x_1). \tag{2}$$

In words, a process is classical if marginalizing over measurement results is identical to not measuring at any given time $t_k$. Since measurements can be disturbing in quantum mechanics, even on average, the validity of Eq. (2) signifies the absence of quantum effects from the perspective of measuring $X$. An example violating Eq. (2) is the famous double slit experiment, see Fig. 1. The following facts further support the idea that this is a good definition of classicality (though, as emphasized above, not the only one).

First, observe that Eq. (2) *defines* a classical stochastic process [39], where it is also known as the *Kolmogorov consistency condition*. Classicality as considered here therefore has a clear operational meaning, which was also used in Ref. [7–15]: *a process is classical if (at least in principle) a classical stochastic process can be used to generate the same measurement statistics.* The idea of defining classicality in this way is rooted in a "black-box-mentality": there might be some very expensive quantum computer in front of you, but if the available measurement statistics can be simulated, or emulated, with a classical stochastic processes, then the measurement statistics *alone* do not allow you to draw the conclusion that there is anything quantum going on in the computer. Furthermore, this definition of classicality also has a clear *practical* motivation because classical stochastic processes are much easier to analyse and simulate than quantum stochastic processes.

Moreover, Eq. (2) implies the validity of Leggett-Garg inequalities [40, 41] and it is closely related but not equivalent to the conditions imposed in the consistent or decoherent histories

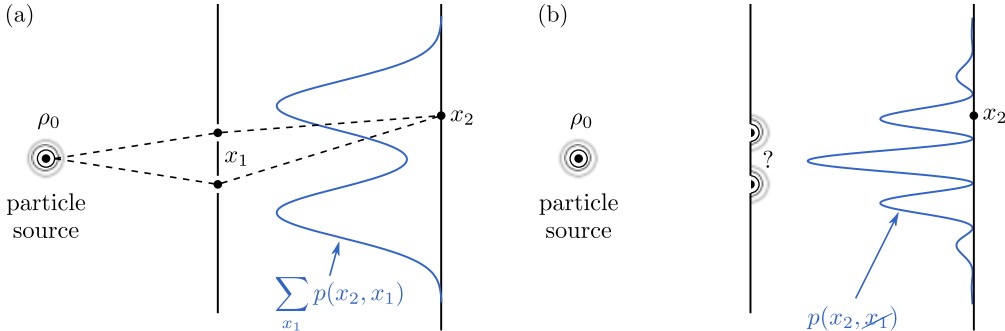

Figure 1: The double slit experiment where a coherent source of particles $\rho_0$ hits a detection screen at position $x_2$ after passing a wall with two holes (the double slit). (a) The particle's location $x_1$ is also measured at the double slit, allowing to decide through which slit it passed (corresponding trajectories indicated by dashed lines). No interference pattern is seen on the detection screen, also not after averaging over $x_1$. (b) There is no measurement of the particle's location at the double slit, it thus retains its coherent wave-like properties and an interference pattern emerges. Clearly, Eq. (2) is violated: the dynamics is non-classical.

formalism [5,6], which we review below in Sec. 2.3. Importantly, however, the definition of classicality used here does not hinge on any specific interpretation of quantum mechanics. Confirming Eq. (2) experimentally only requires measurements of $X$, no further hidden assumption is contained in its definitions. Clearly, classicality is defined with respect to some observable $X$, i.e., a system that behaves classical with respect to $X$ can behave non-classical with respect to a different observable $Y$. Finally, notice that Eq. (2) could be also violated in a *classical* context, for instance, whenever an external agent (e.g., some observer or experimenter) actively intervenes in the process, e.g., by performing feedback control operations [13,14,42]. We exclude these scenarios here by definition of the probabilities in Eq. (1), which in the classical limit (replacing projectors on Hilbert space by characteristic functions on phase space) clearly obey Eq. (2).

In the remainder of this section, we first review the well known decoherence approach and ask whether it explains classicality according to the definition used here (Sec. 2.2). Afterwards, we comment on the relation to the perhaps less well known consistent or decoherent histories approach (Sec. 2.3). Finally, Sec. 2.4 concludes with an intuitive explanation why Eq. (2) can be satisfied for an isolated quantum system.

## 2.2 The decoherence approach for open quantum systems

We consider an open quantum system (OQS) $S$ coupled to some environment or bath $B$. The total Hilbert space is thus a tensor product $\mathcal{H}_S \otimes \mathcal{H}_B$ of the system and bath Hilbert spaces $\mathcal{H}_S$ and $\mathcal{H}_B$, respectively. The dynamics in the full system-bath space is unitary and generated by a Hamiltonian $H_{SB} = H_S + H_B + V_{SB}$ with $H_S$ ($H_B$) the system (bath) Hamiltonian alone (suppressing tensor products with the identity in the notation) and $V_{SB}$ their interaction. The reduced system state $\rho_S(t) = \text{tr}_B\{\rho_{SB}(t)\}$ is obtained from a partial trace of the full system-bath state over the bath degrees of freedom. In contrast to $\rho_{SB}(t)$, $\rho_S(t)$ does *not* evolve in a unitary way.

Decoherence happens whenever it is possible to identify a fixed special basis $\{|\psi_x\rangle\}$ in $\mathcal{H}_S$, which is called the *pointer basis*. The special role of this basis is to ensure that any initial OQS state $\rho_S(0)$ becomes after a characteristic (and typically very short) decoherence time

$t_{\text{dec}}$ diagonal in that basis. In equations,

$$\rho_S(0) = \sum_{x,y} c_{x,y}(0)|\psi_x\rangle\langle\psi_y| \quad \underset{t \geq t_{\text{dec}}}{\longrightarrow} \quad \rho_S(t) \approx \sum_x p_x(t)|\psi_x\rangle\langle\psi_x|. \tag{3}$$

Here, the $c_{x,y}(0) = \langle\psi_x|\rho_S(0)|\psi_y\rangle$ are complex numbers, which ensure positivity and normalization of $\rho_S(0)$ but are otherwise arbitrary, and the $p_x(t)$ are the probabilities to be in state $|\psi_x\rangle$ at time $t$. Equation (3) is indeed a remarkable robust prediction of OQS theory [1–4, 43, 44]. In particular, we repeat that the pointer basis is *fixed*, i.e., it does not depend on the initial system state, but it is determined by the system-bath Hamiltonian and the initial bath state (though the dependence on the latter should be mild in realistic situations). The pointer basis is also often described as "stable", "robust" or "objective" [1–4] and we come back to these properties below.

We further add some clarifications. First, for the sake of generality one should stress that the pointer basis might not be a basis of pure states $|\psi_x\rangle$, but rather a complete set of orthonormal projectors $\{\Pi_x^S\}$ acting on $\mathcal{H}_S$, where certain projectors can have a rank greater than one [45]. In that case, there exist "decoherence-free subspaces" (caused, e.g., by additional conservation laws), but they do not change the fundamental point of our discussion and we continue to call $\{\Pi_x^S\}$ the pointer basis for simplicity. Moreover, we here assume pointer states to be orthogonal, which is typically the case for finite dimensional OQS, but pointer states can be non-orthogonal (e.g., coherent states of a harmonic oscillator [46]). Second, in Eq. (3) we allowed the probabilities $p_x(t)$ to be time-dependent. Their change, however, typically happens on a time scale much slower than the decoherence time scale (see, e.g., Ref. [47]) and is called "dissipation"—a phenomenon already known from classical open systems. Third, we remark that a more nuanced presentation of decoherence is possible. For instance, in order to determine the measurement basis, Zurek in his seminal paper was actually interested in the decoherence of the measurement apparatus, which was in turn coupled to the system to be measured *and* an environment [48]. However, also the measurement apparatus is an OQS, and for the remainder of this paper it is not necessary to explicitly distinguish between system and measurement apparatus. In the following we call the phenenology explained above *OQS decoherence* to distinguish it from the decoherent histories mentioned later.

Next, we ask whether decoherence explains the emergence of classicality according to Eq. (2) if applied to a system observable $X_S = \sum_{x=1}^{M} \lambda_x \Pi_x^S$, i.e., an observable commuting with the pointer basis and acting trivially on the bath space. To this end, we first confirm that for all times larger than the decoherence time $t_{\text{dec}}$ we have

$$\mathcal{D}\rho_S \equiv \sum_x \Pi_x^S \rho_S(t)\Pi_x^S = \rho_S(t), \tag{4}$$

where $\mathcal{D}$ is a *dephasing operation* in the pointer basis.[1] Thus, measuring and averaging is *identical* to not measuring. Next, let us additionally assume that Eq. (4) holds for the full system-bath state:

$$\mathcal{D}\rho_{SB}(t_k) = \sum_x \Pi_x^S \rho_{SB}(t_k)\Pi_x^S = \rho_{SB}(t_k). \tag{5}$$

If that is the case, one can confirm our definition of classicality, i.e., the validity of Eq. (2). However, Eq. (4) *does not imply* Eq. (5), even though the converse is true. Thus, OQS decoherence does not imply classical measurement statistics according to Eq. (2).

It is thus worth thinking about which condition on top of decoherence could imply classicality, and it seems that two different strategies are conceivable.

---

[1]In principle, Eq. (3) implies only an approximate equality ($\approx$) in Eq. (4). However, since classical behaviour should be always understood as some approximation, we replace $\approx$ by $=$ whenever we mean "equal up to some irrelevant measurement error".

The first strategy takes a more detailed look at the *environmental* degrees of freedom. Indeed, the validity of Eq. (5) is equivalent to having vanishing quantum discord [49] in the pointer basis, and testing Eq. (4) has been suggested as a tool to probe non-classical system-bath correlations [50]. However, deciding whether the system-bath state has zero quantum discord or not requires knowledge of the *full* system-bath state. This knowledge is unavailable experimentally and therefore the condition of zero quantum discord is inaccessible from an operational perspective of OQS theory. Moreover, the idea of having zero quantum discord is problematic from the perspective of having a unitarily evolving "universe" consisting of the system and the bath as explained later in Sec. 2.3.

However, a refinement of this idea is possible and has lead to the recently much studied approach of quantum Darwinism, see Ref. [36–38] and references therein. In a nutshell, quantum Darwinism starts by dividing the bath into many different "fragments" $F \subset B$ and asserts that most fragments, even those of small size, have (close) to zero quantum discord with respect to the pointer basis, i.e., Eq. (5) applies to most states $\rho_{SF}(t_k) = \text{tr}_{B \setminus F}\{\rho_{SB}(t_k)\}$, where $B \setminus F$ denotes all bath degrees of freedom except those of the fragment. The resulting classical correlations between the system $S$ and most fragments $F$ allow external observers to learn about the system state even by only looking at a small fragment $F$ of the bath, and different observers looking at different small fragments will agree about the state of $S$. Thus, an objective world emerges.

For the important mechanism of photons scattered off some material object the idea of quantum Darwinism is indeed intuitively appealing because the scattered photons allow different observers, by looking at different narrow angles at the object, to infer, e.g., the same colour or position of it. Moreover, since photons are non-interacting and scatter off to infinity, it becomes clear that their detection does not change the future evolution of the object. As a consequence, Eq. (2) follows.

For other environments, however, the applicability of quantum Darwinism is less clear and whether it guarantees the emergence of classical measurement statistics is unknown at present. In condensed matter and other situations, the bath does not split into non-interacting fragments and perturbations might not be able to escape to infinity. In this case, quantum Darwinism will generically hold at most for transient times [51], yet objectivity and Kolmogorov consistency might nevertheless arise—as this work will indeed confirm now within and later also without OQS decoherence.

Within the paradigm of OQS decoherence, this brings us to the second strategy implying classical behaviour. This strategy is different from the first by rejecting the idea that information about (fragments of) the bath is directly accessible or known. Instead, solely the degrees of freedom of the OQS are deemed operationally accessible, and it then becomes necessary to think about how could one *locally* decide whether the pointer states are stable, robust or objective. Historically, Zurek introduced for this purpose the "predictability sieve" [52], which requires to compute the change in von Neumann entropy of the OQS state $\rho_S(t)$ as a function of a pure initial state $\rho_S(0) = |\psi(0)\rangle\langle\psi(0)|_S$. If it changes very slowly, the dynamics are predictable as the state remains approximately pure, but if it changes very rapidly, the dynamics are unpredictable as the state becomes very mixed. Now, from what we said above, we see that initial pointer states are characterized by a slow change in von Neumann entropy, whereas superpositions of pointer states quickly decohere into a mixture on a time scale $t_{\text{dec}}$. The predictability sieve thus selects out the pointer states.

Although the predictability sieve has appealing properties, it is ultimately not satisfactory for the following reason. If we want to find out whether something is predictable (or stable, robust or objective), it is best to really "take a look at it". For instance, the memories in our computers are stable because we can *repeatedly* read them out without changing their state. How can this idea be formalized mathematically? Clearly, one way to test this property is to

measure the OQS in the pointer basis, say at some time $t_1 \geq t_{\mathrm{dec}}$, and then to look whether this measurement influences the future evolution of the OQS at some time $t_2 > t_1$, for instance, by checking whether the future probabilities of the pointer states depend on the measurement at time $t_1$. We now notice that this idea to check for predictability is *exactly equal to testing* our definition of classicality in Eq. (2) for the pointer basis $\{\Pi_x^S\}$. Clearly, other ways are possible, but within this second strategy they should always be related to watching the response to some form of external perturbation or intervention on the OQS: do the pointer states remain stable if we shake them a bit?

After having spelled out the basic idea, it remains mostly a technical problem to realize that OQS decoherence in the form of Eq. (3) *plus* the condition of *Markovianity* as defined in Ref. [16] (for introductions see Refs. [13, 14]) is sufficient to imply classical measurement statistics. In short, this definition of Markovianity is based on the idea that local operations on the system performed by an external agent do not influence the OQS dynamics generated by the environment. Importantly, the property of Markovianity can be checked by local system operations only (no knowledge of the bath state is required) [13, 14, 16, 17]. However, since the definition of Markovianity requires to check multi-time correlations (in complete analogy to the classical definition), knowledge of the time evolution of $\rho_S(t)$ alone is insufficient to check for Markovianity (a discussion focused on this point can be found in Ref. [18]).

The connection to Markovianity now becomes transparent by realizing that "shaking a bit the system" is an external intervention that will sooner or later also influence the environment. Can this influence of the environment cause a different behaviour of the system? If the answer is no, then this precisely means that the dynamics is Markovian. In that case, we can conclude the following. First, we found above that OQS decoherence implies that $\mathcal{D}\rho_S(t_1) = \rho_S(t_1)$ for $t_1 \geq t_{\mathrm{dec}}$. Obviously, one also has $\mathcal{I}\rho_S(t_1) = \rho_S(t_1)$ where $\mathcal{I}$ is the identity operation which, operationally speaking, literally means "do nothing!" Now, according to the definition of Markovianity explained above, the dynamics induced by the environment is insensitive to local operations on the system performed by an external agent. Since the two operations $\mathcal{D}$ and $\mathcal{I}$ do not change the OQS state, the future dynamics is insensitive to the dephasing operations, that is: the pointer states are stable and the dynamics is classical. Formal definitions and a proof are given in Appendix A.1.

This important message together with various other notions (some of which are only introduced in Sec. 2.3) is summarized in Fig. 2. Notably, the role of multi-time statistics to probe the stability of pointer states and its connection to Markovianity is not the focus of the OQS decoherence approach [1–4] and also not of quantum Darwinism [36–38], although — within the histories framework introduced below — connections (with varying degree of generality and rigour) have beed made [53–56]. Moreover, a clear-cut consensus from numerical studies about the relation between non-Markovianity and quantum Darwinism has not yet emerged [57–59]. Thus, it seems worthwhile in the future to look for a closer connection of (non-)Markovianity and quantum Darwinism in physical relevant situations.

Finally, we make two more important observations. First of all, the question "what is the pointer basis?" is non-trivial and has been only answered in certain limiting cases (e.g., very strong or very weak system-bath coupling) [1–4]. In general, for a complex open many-body system coupled to a complex many-body environment the pointer basis is not known. It is an advantage of the approach presented in Secs. 2.4 and 3 of this paper that no pointer basis needs to be identified. Second, all what we said above was restricted to the OQS paradigm, i.e., local observables defined on a system-bath tensor product structure, whose identification can be non-trivial [60]. This restriction is also lifted in the present approach, which makes it appealing for questions usually studied within the formalism reviewed next.

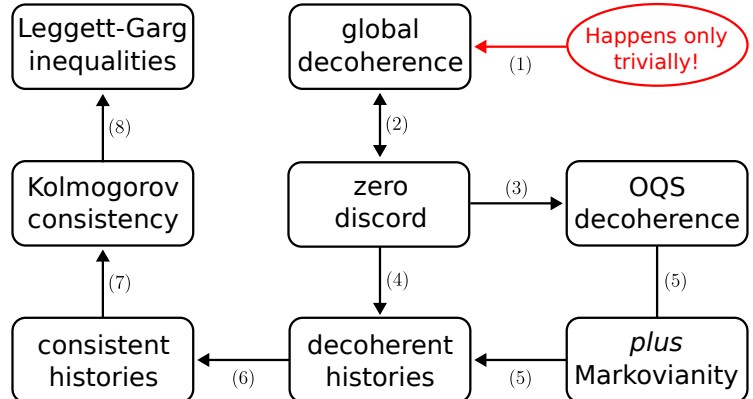

Figure 2: Overview over relations between different concepts defined in the text. Arrows mean strict mathematical implications (only for arrow (7) it is currently not known whether also the inverse implication holds). Comments: (1) is explained at the end of Sec. 2.3 (and "happens only trivially" is the only term not defined rigorously in this diagram), (2) assumes a local system observable (discord is undefined without a system-bath tensor product structure) and is here understood as applying *repeatedly* for all times $t_k$ considered in Eq. (2), (3) is discussed around Eq. (5), (4) is easily shown with the definition in Sec. 2.3, (5) is explained in Sec. 2.2 and proven in Sec. A.1 (the strict implication follows from the existence of classical processes that are non-Markovian), (6) and (7) are shown in Sec. 2.3, (8) follows because the derivation of Leggett-Garg inequalities assumes Kolmogorov consistency [40, 41].

## 2.3 Consistent and decoherent histories

The consistent or decoherent histories formalism is an attempt to explain how standard reasoning based on classical logic can be applied in an isolated quantum system in general, and in the cosmological Universe in particular [5, 6, 61–66]. As we will see, it is closely related to the Kolmogorov consistency criterion. It has been also viewed as a new *interpretation* of quantum mechanics [6, 61, 63], but this view has been fiercely debated [67]. We here prefer to remain agnostic about this question, but simply point out again that the mathematical definitions and relations introduced below make sense without reference to any particular interpretation of quantum mechanics.

The approach starts by introducing a *decoherence functional* $\mathfrak{D}$ for two histories $\mathbf{x} \equiv (x_n, \ldots, x_2, x_1)$ and $\mathbf{y} \equiv (y_n, \ldots, y_2, y_1)$ "happening" at times $t_n > \cdots > t_1$:

$$\mathfrak{D}(\mathbf{x}; \mathbf{y}) \equiv \mathrm{tr} \left\{ \Pi_{x_n} U_n \ldots \Pi_{x_2} U_2 \Pi_{x_1} U_1 \rho_0 U_1^\dagger \Pi_{y_1} U_2^\dagger \Pi_{y_2} \ldots U_n^\dagger \Pi_{y_n} \right\}. \tag{6}$$

Here, the projectors and unitary time evolution operators have the same meaning as in Eq. (1) and we immediately confirm that the diagonal elements of the decoherence functional correspond to our previously introduced joint probabilities: $p(x_n, \ldots, x_1) = \mathfrak{D}(\mathbf{x}; \mathbf{x})$.

Depending on the precise reference, different notions of "consistency", "decoherence" or "(quasi)classicality" have been introduced based on the decoherence functional. It is beyond the scope of this article to review them all here, so we restrict the discussion to the two most commonly employed definitions. First, Griffith originally proposed what we here call the *consistent histories* condition [61]:

$$\text{consistent histories:} \quad \Re[\mathfrak{D}(\mathbf{x}; \mathbf{y})] = 0, \quad \text{for all} \quad \mathbf{x} \neq \mathbf{y}, \tag{7}$$

i.e., the vanishing of the real part of the decoherence functional for different histories. Gell-Mann and Hartle, among others (see, e.g., Refs. [65, 66] and references therein) prefer to use

the following condition, which we call the *decoherent histories* condition:

$$\text{decoherent histories:}\quad \mathfrak{D}(\mathbf{x};\mathbf{y}) = 0\,, \quad \text{for all}\quad \mathbf{x} \neq \mathbf{y}\,. \tag{8}$$

Three immediately obvious remarks follow. First, condition (8) implies Eq. (7). Second, $\mathfrak{D}(\mathbf{x};\mathbf{y}) = 0$ always if $x_n \neq y_n$, i.e., the final "measurement results" cannot be different. Third, Eq. (7) implies the Kolmogorov consistency condition (2) (and hence so does Eq. (8)). Confirming the last result requires a few lines of algebra, but it was already shown by Griffiths [61] and many others and will thus not be repeated here.

A not so obvious conclusion is that the decoherent histories condition is *strictly* stronger than the consistent histories condition. This is explained with a result of Diósi [68], who considered two decoupled quantum systems $A$ and $B$ prepared in a decorrelated state $\rho_A(t_0) \otimes \rho_B(t_0)$, unitarily evolving without interaction according to $U_A \otimes U_B$ and measured with decorrelated projectors $\Pi_x^A \otimes \Pi_{x'}^B$. In this situation, one immediately confirms that the joint decoherence functional factorizes as $\mathfrak{D}_{AB}(\mathbf{x},\mathbf{x}';\mathbf{y},\mathbf{y}') = \mathfrak{D}_A(\mathbf{x};\mathbf{y})\mathfrak{D}_B(\mathbf{x}';\mathbf{y}')$, where unprimed (primed) histories refer to subsystem $A$ ($B$). Now, suppose that $A$ and $B$ separately satisfy the decoherent histories condition. Then, this is also the case for the non-interacting composite $AB$, as one would intuitively expect. However, this conclusion does not hold for the consistent histories condition, thus the latter cannot imply the former. Thus, Diósi's argument is typically invoked to say that Eq. (8) is a more meaningful condition than Eq. (7).

Now, if we consider the probabillities for such a decoupled system, they factorize as expected: $p_{AB}(\mathbf{x},\mathbf{x}') = p_A(\mathbf{x})p_B(\mathbf{x}')$. Interestingly, if $A$ and $B$ separately satisfy the Kolmogorov consistency condition, then this is also true for the composite $AB$. Thus, Diósi's argument cannot be invoked to refute our definition of classicality based on Eq. (2).[2] Two further statements are noteworthy: First, what we said above implies that the decoherent histories condition is strictly stronger than the Kolmogorov consistency condition. Second, confirming the decoherent histories condition experimentally is obviously much harder than confirming the Kolmogorov consistency condition.

Next, we turn to the relation between decoherent histories and OQS decoherence, which obviously has been already the topic of previous works, see, e.g., the above references and in particular also Refs. [69, 70]. Quite intuitively, one would expect that histories defined by measurements in the pointer basis naturally satisfy the decoherent histories condition, i.e., that OQS decoherence generates decoherent histories, but it has been recognized that this relation is not that easy [69, 70]. Indeed, since OQS decoherence alone is not sufficient to imply Kolmogorov consistency, it also cannot be sufficient to imply decoherent histories. Interestingly, the extra condition of Markovianity is again sufficient to show that OQS decoherence implies decoherent histories. The proof of this statement is given in Appendix A.1 (see also Refs. [53–56] for related work).

Finally, we turn to the question whether decoherence in a stronger, global sense can explain the emergence of decoherent histories. Here, *global decoherence* means that the unitarily evolving state is block diagonal with respect to the projectors $\{\Pi_x\}$, i.e., $\rho(t) = \sum_x \Pi_x \rho(t)\Pi_x$ (note that this implies zero quantum discord, Eq. (5), for local system projectors). If that is the case for all times $t_k$ appearing in definition (6) of the decoherence functional, one immediately confirms that the histories satisfy the decoherent histories condition. Unfortunately, however, if $\rho(t)$ is unitarily evolving this can in general only be the case for trivial situations. To see this, we restrict the discussion to pure states $|\psi(t)\rangle$, which is sufficient for isolated systems. Now, from $\sum_x \Pi_x = I$ we infer that every pure state can be written as $|\psi(t)\rangle = \sum_x \sqrt{p_x(t)}e^{i\varphi_x(t)}|\psi_x(t)\rangle$ with $p_x(t)$ the probability to measure outcome $x$ and $\Pi_y|\psi_x(t)\rangle = \delta_{x,y}|\psi_x(t)\rangle$. Next, notice that the only states $|\psi(t)\rangle$ that are block diagonal or

---

[2]Diósi also gives a second argument to argue in favour of decoherent instead of consistent histories. Again, also this second argument does not disfavour our definition.

globally decohered are states with $p_x(t) = \delta_{x,x^*}$ for some $x^*$, i.e., these states are fully localized in one subspace or, with respect to the measurement outcomes, we can say that they are *deterministic*. Now, this can certainly happen in some cases, for instance, if the dimension of the subspace $x^*$ dominates by far all other subspaces (which corresponds to the usual criterion of $x^*$ describing an equilibrium state in statistical mechanics), or if the times $t_k$ are carefully chosen such that $|\psi_x(t_k)\rangle$ is localized in one subspace. Moreover, if $\Pi_x$ commutes with the Hamiltonian, its probability remains constant and always generates classical statistics.

However, excluding globally conserved quantities, considering interesting nonequilibrium dynamics and rejecting the unrealistic idea that we are able to carefully choose the times $t_k$, the state $|\psi(t)\rangle$ cannot remain block diagonal. For instance, if the state has a high initial probability $p(x_0) \lesssim 1$ for some $x_0$ and a high probablity for some final state $p(x_n) \lesssim 1$ with $x_n \neq x_0$, then there must be some intermediate time $t$ where the state passed from $x_0$ to $x_n$ such that $p_{x_0}(t) = 1/2$. Thus, global decoherence can only happen under trivial or unrealistic circumstances.

A summary of this and the last section can be found in Fig. 2.

## 2.4 The new approach: General picture

We now discuss the general picture behind the new approach from Refs. [7,8,15]. It is claimed to be "new" for two reasons. First, as discussed above, it defines classicality in terms of the Kolmogorov consistency condition. This differs from the basic question in OQS decoherence ("What is the measurement/pointer basis?") and it is close to but still different from the histories approach. In particular, Kolmogorov consistency can be independently well motivated by asking the question "When can a quantum process be simulated by a classical process?", as also done recently in Refs. [9–12]. Second, emphasis is put on the following two physical aspects. First, the focus is *not* on OQS: even global observables of an isolated system can behave classical. Second, *non-integrability* is regarded as essential, or at least very helpful, to derive classicality. While the relation to chaos has been also studied in OQS decoherence (see, e.g., Refs. [71–79]), it has not been regarded as essential: the traditional workhorse model of OQS theory uses an integrable bath of harmonic oscillators ("Caldeira-Leggett model") prepared in a canonical Gibbs ensemble. This is avoided in the following by considering pure states.

To the best of the author's knowledge, the basic physical picture behind this emergence of classicality has been already explained by van Kampen in 1954 [80] (without, however, receiving any attention of the community working on the quantum-to-classical transition). Three basic ingredients, which can hardly count as assumptions, plus one major assumption are necessary to see the emergence of classicality. The three ingredients are: (i) the system has a well-defined overall energy, i.e., the energy spread $\Delta E$ of the initial wavefunction is sufficiently narrow;[3] (ii) the system has many particles $N \gg 1$, i.e., the Hilbert space dimension of the aforementioned energy shell is exponentially large: $D \equiv \dim \mathcal{H} \sim \exp(N)$; (iii) the system is non-integrable or, more precisely, it should obey the *eigenstate thermalization hypothesis* (ETH). Given the success of the ETH this is considered a mild assumption for realistic many-body systems found in nature [23, 24].

The major assumption concerns the observable $X$ that one is probing: according to Refs. [15, 80] it should be *coarse* and *slow*. Coarseness means that the number of potential measurement outcomes is much smaller than the Hilbert space dimension: $M \ll D$. Again, this can hardly count as an assumption. In particular, observe that an observable $X_S$ defined for an OQS is a coarse observable in the full system-bath space. Slowness instead is the *crucial* assumption and it has been discussed in detail (together with various subtleties) in Ref. [15].

---

[3]Recall that energy is conserved in an isolated system. So if the initial state is a superposition of *macroscopically different* energies, the analysis should be carried out separately for each component. Moreover, if there are further conserved quantities, the same argument has to be also applied to them.

Intuitively, it means that the time scale $\tau_X$ on which $\langle X \rangle(t) = \mathrm{tr}\{X U_t \rho_0 U_t^\dagger\}$ evolves is much longer than the microscopic evolution time scale $\hbar/\Delta E$. This is equivalent to saying that the matrix with elements $X_{km} = \langle k|X|m \rangle$ with respect to an ordered energy eigenbasis $\{|k\rangle\}$ is *narrowly banded*.

It is interesting to contrast this approach to previous work done within the consistent or decoherent histories formalism, where slow (or quasi-conserved) observables also played an essential role [63,65,66,81]. Without noting the work of von Kampen, the focus in these works was to derive the consistent or decoherent histories condition by arguing that the wave packet remains strongly localized along some trajectory, which is described by a classical deterministic equation, i.e., it was argued that the wave function $|\psi(t)\rangle$ should remain approximately localized in one (time-dependent) subspace $\Pi_{x(t)}$ throughout the dynamics.

It is questionable whether this is always an adequate idea, but, importantly, the assumption of remaining localized around some classical trajectory is also not necessary.

As it will come clear below, the pure state $|\psi(t)\rangle$ is allowed to have an abundance of coherences (even maximal coherences) and can still behave classically. This marks another and perhaps the most important novel aspect.

It is also interesting to connect the assumptions above to the OQS decoherence approach. To this end, consider an OQS and let $X = H_S$ be the system Hamiltonian. Since $H_S$ is locally conserved ($[H_S, H_S] = 0$), $H_S$ is a slow observable provided that the coupling $V_{SB}$ is weak enough. Furthermore, it is also coarse since $1 \ll \dim \mathcal{H}_B$. Thus, in the weak coupling regime local measurements of the energy should give rise to classical statistics obeying the Kolmogorov consistency condition (2). This is unison with the predictions of the pointer basis in the decoherence approach, but we repeat that the identification of a pointer basis is not necessary in the present approach. However, it should be emphasized that the notion of slowness is subtle and it does not seem to be a sufficient criterion for classicality: by precisely tuning the "fine-structure" of $X_{km}$ it appears that one can generate arbitrary exceptions to the "rule" [82], albeit those might not be generic. In any case, it provides a different perspective on the problem and it gives an immediate intuitive explanation why the world around us appears classical: human senses are simply to slow and coarse to resolve the evolution of fast observables that could potentially show quantum effects.

So how can it be that decoherence is not necessary to generate classical measurement statistics? The following picture lacks rigour, but gives some intuition.

To start with consider a two-level system with Hamiltonian $\frac{\Delta}{2}\sigma_z = \frac{\Delta}{2}(|1\rangle\langle 1| - |0\rangle\langle 0|)$ and as the observable choose $X = \sigma_x$. Moreover, let the initial state $\rho_0$ with respect to the eigenbasis $|\pm\rangle = (|0\rangle \pm |1\rangle)/\sqrt{2}$ of $\sigma_x$ be parametrized as

$$\rho_0 = \begin{pmatrix} (1+\delta)/2 & re^{i\phi} \\ re^{-i\phi} & (1-\delta)/2 \end{pmatrix}, \quad \delta \in [-1,+1], \quad 0 \leq r \leq \frac{\sqrt{1-\delta^2}}{2}, \quad \phi \in [0, 2\pi). \quad (9)$$

Here, the parameter $r$ quantifies the "strength" of the coherences in the $\sigma_x$ basis, which is always upper bounded by $\sqrt{1-\delta^2}/2$ due to the positivity requirement $\rho_0 \geq 0$. Now, at an arbitrary time $t$ the system state in the same basis reads

$$\rho_t = \begin{pmatrix} (1+\delta\cos\Delta t)/2 + r\sin\phi\sin\Delta t & r(\cos\phi + i\sin\phi\cos\Delta t) - \frac{i\delta}{2}\sin\Delta t \\ r(\cos\phi - i\sin\phi\cos\Delta t) + \frac{i\delta}{2}\sin\Delta t & (1-\delta\cos\Delta t)/2 - r\sin\phi\sin\Delta t \end{pmatrix}. \quad (10)$$

The diagonal elements equal the probabilities to measure spin up $|+\rangle$ or spin down $|-\rangle$ with respect to the $x$-direction. Their time evolution is strongly influenced by the coherences and therefore the dynamics is not classical. Of course, this is not a counterexample since the system is neither a non-integrable many-body system nor is the observable coarse and slow.

So what happens for a coarse observable in a non-integrable many-body system? The single elements of Eq. (10) now become blocks of many elements and the probability $p_x(t)$

to find the system in some state $x$ becomes the trace over block $x$. It will typically contain a sum of contributions from many coherences $\langle i|\rho_0|j\rangle = r_{ij}e^{i\phi_{ij}}$ of the initial state, schematically written as:

$$p_x(t) \sim \sum_{i,j} r_{ij} \sin\phi_{ij} \sin\Delta_{ij} t \,. \tag{11}$$

Now, observe the following facts. First, for a coarse observable of a many-body system the number of terms contributing to the sum is *huge* (of the order $e^N$ with $N$ the particle number). Second, for a non-integrable system the energy differences $\Delta_{ij}$ are incommensurable (apart from rare accidental degeneracies) and effectively random. Thus, unless the $\phi_{ij}$ are precisely tuned or $r_{ij} = 0$ for most but a few pairs $(i,j)$, Eq. (11) is a sum of many terms of random sign and small magnitude.[4] Thus, the enormous amount of coherences *cannot add up* to a significant contribution and therefore it effectively does not matter whether coherences are present or not, i.e., as long as one only asks questions about the measurement statistics in Eq. (1) we can set $r_{ij} = 0$ for all $(i,j)$.

This explanation for classical behaviour is essentially statistical and similar in spirit to the explanation of the second law. Yes, it is possible that the positions and momenta of all the molecules in the air surrounding you conspire such that they can be all found in one corner of the room in the next second, yet this possibility is *extremely unlikely*. Similarly, it is possible that all microscopic coherences of a coarse observable align in phase to give rise to a strong contribution and, consequently, a strong violation of Kolmogorov consistency, yet this is again extremely unlikely. In essence, this also underlies the decoherence approach. Yes, it is possible that a qubit in contact with a bath suddenly "recoheres", yet this would require again a very unlikely because precisely tuned cooperation of many phases of the system-bath state. Thus, the emergence of classical behaviour is related to the general phenomenon of *irreversibility*, which is extremely hard to avoid given our coarse human senses.

Finally, one might wonder where above the assumption of *slowness* enters. This is indeed not directly visible here. However, our argument why Eq. (11) is small was based on assumptions about the number of coherences $r_{ij}$ and the absence of correlations within and between the coherences $r_{ij}$, phases $\phi_{ij}$ and frequencies $\Delta_{ij}$. Unfortunately, and in particular for states prepared out of equilibrium, these assumptions become questionable. Indeed, a state drawn at random can be shown to give with overwhelming probability rise to equilibrium statistics [83–86]. Nonequilibrium states thus cannot be completely random and must possess a sufficient amount of correlations. Slowness now helps because the microscopic state evolves on a much shorter time scale than the observable, effectively randomizing many phases before any noticeable change in $p_x(t)$ occurs. Thus, from the perspective of the slow observable $X$, the systems looks *locally equilibrated* and the precise microstate no longer matters [15, 80].

## 3 Classicality: Derivation and numerical verification

### 3.1 Derivation using random matrix theory

To show the approximate validity of the Kolmogorov consistency condition (2) one needs a model that, ideally, is as general as possible to cover a wide range of scenarios while being at the same time also specific enough to permit explicit calculations. Unfortunately, these two desiderata are often mutually exclusive. In Ref. [15] the model was assumed to obey the ETH, which is currently considered to be a mild assumption for most realistic many-body systems

---

[4]Recall that $r_{ij}^2 \le p_i p_j$ (by Cauchy-Schwarz) where $p_i$ is the probability to find the system in a certain microscopic state $|i\rangle$. Generically, a system has overlap with an enormous amount of microscopic states and, since $\sum_i p_i = 1$, this implies that $r_{ij}$ must be very small.

found in nature [23, 24]. The drawback of this generality was that some plausible but at the end unproven assumptions entered the derivation.

Here, a random matrix theory approach is used, which has been successful in modelling a variety of generic properties of complex systems [19–24]. Indeed, our current understanding of the ETH is much based on random matrix theory although the ETH has been shown to be valid for a much larger class of models. The model considered here is therefore more restrictive than the model of Ref. [15], but it comes with the benefit that we need less unproven assumptions (albeit we still need some).

To capture the relevant physics of a non-integrable many-body system we follow Deutsch [87] and others [88–96] and consider a Hamiltonian of the form

$$H = H_0 + \epsilon H_1 \,. \tag{12}$$

Here, $H_0$ is some "baseline" Hamiltonian, $\epsilon$ a small parameter and $H_1$ a banded random Hermitian matrix chosen, e.g., from the Gaussian orthogonal or unitary ensemble. For instance, $H_0 = H_S + H_B$ could be the bare system and bath Hamiltonian and $\epsilon H_1 = V_{SB}$ their (weak) interaction, but many more examples are imaginable. Note that $H_0$ does not need to be integrable, it is only assumed to describe a many-body system with an extremely dense spectrum in the considered energy interval (recall ingredients (i) and (ii) in Sec. 2.4). Moreover, the model does not literally assume that the perturbation is random. Instead, the basic idea of random matrix theory is that some property holds for the overwhelming majority of random perturbations and that the real physical (and non-random) Hamiltonian then also belongs to this overwhelming majority. Finally, the smallness of $\epsilon$ implies that the range of the spectrum and the mean level spacing $\delta e$ of $H_0$ and $H$ are comparable, but their eigenvectors are still strongly perturbed as long as $\epsilon$ is larger than the extremely small level spacing $\delta e$.

Let $|\mu\rangle$ and $|m\rangle$ be the eigenvectors of $H_0$ and $H$, respectively. A central role in the following is played by the unitary matrix

$$V_\mu^m \equiv \langle m|\mu\rangle \,, \tag{13}$$

which transforms between the eigenbasis of $H_0$ and $H$ and quantifies their overlap. Denoting by $\mathbb{E}[\dots]$ averages over the random matrix ensemble, we use that

$$u(m,\mu) \equiv \mathbb{E}\left[\left|V_\mu^m\right|^2\right] = u(m-\mu), \quad \sum_n u(n) = 1, \quad \max_n u(n) = \mathcal{O}\left(e^{-N}\right), \tag{14}$$

which holds for both the Gaussian orthogonal or unitary ensemble (and even beyond strict Gaussianity) and whose detailed justification is left to the literature [87–99]. The important point is that the overlap between eigenvectors of $H$ and $H_0$ is exponentially small in the particle number $N$ and for our estimate below we will set for notational simplicity $\max_n u(n) = D^{-1}$ with $D$ the Hilbert space dimension of the energy shell (which is exponentially large in $N$).

Next, as an observable we allow any coarse Hermitian operator $X = \sum_{x=1}^M \lambda_x \Pi_x$ that commutes with the unperturbed Hamiltonian, $[H_0, X] = 0$, but not with the perturbation $H_1$ (the case $[H_1, X] = 0$ trivially gives rise to classical dynamics for $X$). Coarseness means that $M \ll D$ and the smallness of $\epsilon$ implies that $X$ evolves on a slow time scale. Note that also observables $X$ with $[H_0, X] \neq 0$ can behave classical [7, 8, 15], but the above assumption turns out to be very convenient for the calculation below.

Then, we consider the joint probability

$$p(x_2, x_1) \equiv \text{tr}\left\{\Pi_{x_2} U_2 \Pi_{x_1} U_1 \rho_0 U_1^\dagger \Pi_{x_1} U_2^\dagger\right\} \tag{15}$$

to measure $x_1$ at time $t_1$ and $x_2$ at time $t_2$ given an arbitrary initial state (perhaps far from equilibrium) $\rho_0$. We further introduce the single time probability

$$p(x_2, \cancel{x_1}) \equiv \text{tr}\left\{\Pi_{x_2} U_2 U_1 \rho_0 U_1^\dagger U_2^\dagger\right\} \,, \tag{16}$$

and consider the difference

$$Q \equiv p(x_2, \cancel{x_1}) - \sum_{x_1} p(x_2, x_1) = \sum_{x_1 \neq y_1} \mathrm{tr}\{\Pi_{x_2} U_2 \Pi_{x_1} U_1 \rho_0 U_1^\dagger \Pi_{y_1} U_2^\dagger\} \in [-1, 1]. \tag{17}$$

The goal in the following is to show that $Q$ is extremely small. This then implies that the Kolmogorov consistency condition (2) is satisfied at the level of arbitrary two-time probabilities given any initial state $\rho_0$. An extension of the derivation to arbitrary $n$-time probabilities with $n > 2$ is certainly desirable, but it is a very complicated problem, likely requiring novel techniques. However, abstracting from Refs. [100–103], where theorems about $n$-time correlations functions for large $n$ were proven under different circumstances, it appears likely that approximate consistency continues to hold also for $n > 2$.

To make analytical progress, we first need some assumption about the initial state $\rho_0$ and at this stage we follow Ref. [15]. In summary, Ref. [15] has described the initial state by a *preparation procedure* using some completely positive map $\mathcal{M}$ such that $\rho_0 = \mathcal{M}\psi_0 = \sum_\alpha K_\alpha \psi_0 K_\alpha^\dagger$, where $\psi_0$ is the state *prior* to the preparation. So far, this is completely general [13, 14, 104], but now two assumptions are introduced. First, by polar decomposition we write $K_\alpha = \sqrt{P_\alpha} V_\alpha$, where $V_\alpha$ is a unitary and $P_\alpha$ a positive operator. It was now assumed that the $P_\alpha = P_\alpha(X)$ are functionally dependent on $X$. Translated into an experimental context, this means that the experimentalist has control over $X$ (for instance, by measuring it), but they are not in control over the precise microstate within the subspaces of $\Pi_x$. Second, it was assumed that the prior state $\psi_0$ is at equilibrium or, technically speaking, Haar randomly distributed in the energy shell. Both assumptions thus express the idea that the initial state preparation can bring a system, which is at equilibrium from a macroscopic point of view (note that $\psi_0$ is a pure state), arbitrarily far from equilibrium *with respect to $X$*. Then, using a measure concentration inequality in form of Levy's lemma [86], it was shown that smallness of Eq. (17) is (in almost all cases) equivalent to showing smallness of

$$q(x_2, x_0) \equiv \frac{1}{D_{x_0}} \sum_{x_1 \neq y_1} \mathrm{tr}\left\{\Pi_{x_2} U_2 \Pi_{x_1} U_1 \Pi_{x_0} U_1^\dagger \Pi_{y_1} U_2^\dagger\right\}, \quad \text{for} \quad x_2 \neq x_0. \tag{18}$$

Here, $D_{x_0} \equiv \mathrm{tr}\{\Pi_{x_0}\}$ is the dimension of the subspace associated to measurement outcome $x_0$. Thus, in essence the term $Q$, which is a *three*-point correlation function for the projectors $\Pi_x$ with *unknown* correlations with respect to the initial state $\rho_0$, got transformed into the term $q(x_2, x_0)$, which is a *four*-point correlation function for the projectors $\Pi_x$ *without* any initial state dependence.

We evaluate Eq. (18) in the energy eigenbasis of $H$ and introduce the following convention, which is perhaps unconventional but useful for later considerations. Since there will be many terms indexed by many quantities $m_1, m_2, \ldots$ and $\mu_1, \mu_2, \ldots$, where $m_i$ ($\mu_i$) labels energy eigenvalues of $H$ ($H_0$), we decide to write labels $m_i$ ($\mu_i$) as superscripts (subscripts) as in Eq. (13) and take the freedom to simply replace them by the number $i$ whenever appropriate. Thus, Eq. (18) then becomes

$$q(x_2, x_0) = \frac{1}{D_{x_0}} \sum_{x_1 \neq y_1} \sum^{1,2,3,4} e^{i\omega^{12}(t_2 - t_1)} e^{i\omega^{43} t_1} \Pi_{x_2}^{12} \Pi_{x_1}^{23} \Pi_{x_0}^{34} \Pi_{y_1}^{41}, \tag{19}$$

where $\omega^{12} = E^1 - E^2$ denotes the difference between two eigenenergies of $H$. Next, we recall that $X$ is a narrowly banded operator (due to its slowness) and this also implies that $\Pi_x$ is narrowly banded (due to the coarseness of $X$) [15]. Thus, in an ordered energy eigenbasis we can safely assume $\Pi_x^{mn} = 0$ if $m - n \geq d$ for a sufficiently large number $d$. Importantly, while $d \gg 1$ can be enormous in realistic applications, a central feature of slowness and coarseness is that still $d \ll D$. Thus, $d/D$ serves as a small parameter in the following and the corresponding

*restricted* summation is denoted as $\sum^{1\approx2\approx3\approx4}$. Finally, notice that $q(x_2,x_0)=0$ if $t_2=t_1$ or $t_1=0$, which allows to turn Eq. (19) into

$$q(x_2,x_0)=\frac{1}{D_{x_0}}\sum_{x_1\neq y_1}\sum^{1\approx2\approx3\approx4}(e^{i\omega^{12}(t_2-t_1)}-1)(e^{i\omega^{43}t_1}-1)\Pi^{12}_{x_2}\Pi^{23}_{x_1}\Pi^{34}_{x_0}\Pi^{41}_{y_1}. \tag{20}$$

We continue by using Eq. (13) and $[H_0,X]=0$ to obtain

$$\Pi^{mn}_x=\langle m|\Pi_x|n\rangle=\sum_{\mu,\nu}\langle m|\mu\rangle\langle\mu|\Pi_x|\nu\rangle\langle\nu|n\rangle=\sum_{\mu}\chi_{\mu}(x)V^m_{\mu}\bar{V}^n_{\mu}. \tag{21}$$

Here, $\chi_{\mu}(x)$ is the indicator function which is one if and only if $\Pi_x|\mu\rangle=|\mu\rangle$ and zero otherwise. Furthermore, note that we use an overbar to denote the complex conjugate. Inserting Eq. (21) into Eq. (20), we arrive at

$$\begin{aligned}q(x_2,x_0)=\frac{1}{D_{x_0}}\sum_{x_1\neq y_1}\sum^{1\approx2\approx3\approx4}_{1,2,3,4}(e^{i\omega^{12}(t_2-t_1)}-1)(e^{i\omega^{43}t_1}-1)\\ \times\chi_1(x_1)\chi_2(y_1)\chi_3(x_0)\chi_4(x_2)V^1_4\bar{V}^2_4V^2_1\bar{V}^3_1V^3_3\bar{V}^4_3V^4_2\bar{V}^1_2.\end{aligned} \tag{22}$$

We have now reached a point, where we can try to evaluate $q(x_2,x_0)$ using random matrix theory. However, since $q(x_2,x_0)\in\mathbb{R}$ can be positive or negative, showing smallness of $q(x_2,x_0)$ on average is only an indicator, but not a gurantee that $q(x_2,x_0)$ is small in general (because it could also strongly fluctuate for different realizations of the random Hamiltonian). Thus, we will actually show that $[q(x_2,x_0)]^2$ is small, which establishes smallness of $q(x_2,x_0)$ *and* its variance, and which is one point where we go beyond the treatment of Ref. [15]. Thus, we aim to evaluate

$$\mathbb{E}\left\{[q(x_2,x_0)]^2\right\}\approx \tag{23}$$
$$\frac{1}{D^2_{x_0}}\sum_{x_1\neq y_1}\sum_{x'_1\neq y'_1}\sum^{1\approx2\approx3\approx4}_{1,2,3,4}\sum^{5\approx6\approx7\approx8}_{5,6,7,8}(e^{i\omega^{12}(t_2-t_1)}-1)(e^{i\omega^{43}t_1}-1)(e^{i\omega^{56}(t_2-t_1)}-1)(e^{i\omega^{87}t_1}-1)$$
$$\times\chi_1(x_1)\chi_2(y_1)\chi_3(x_0)\chi_4(x_2)\chi_5(x'_1)\chi_6(y'_1)\chi_7(x_0)\chi_8(x_2)$$
$$\times\mathbb{E}\left[V^1_4\bar{V}^2_4V^2_1\bar{V}^3_1V^3_3\bar{V}^4_3V^4_2\bar{V}^1_2V^5_8\bar{V}^6_8V^6_5\bar{V}^7_5V^7_7\bar{V}^8_7V^8_6\bar{V}^5_6\right].$$

Note that the ensemble average is only performed over the matrix elements $V^m_{\mu}$, but excludes the frequencies $\omega^{mn}$. In principle, these should be included in the ensemble average as well, but the smallness of the random perturbation and the extremely small mean energy level spacing $\delta e$ suggest that the behaviour of $q(x_2,x_0)$ is insensitive to small perturbations of $\omega^{mn}$ for times much smaller than the extremely long Heisenberg time $\hbar/\delta e$ (for further justification see Refs. [93–95, 105]).

Evaluation of Eq. (23) is facilitated by the fact that ten constraints apply. First, due to the factors $e^{i\omega^{ij}t}-1$ we infer the four constraints $m_1\neq m_2$, $m_3\neq m_4$, $m_5\neq m_6$ and $m_7\neq m_8$. Second, due to the fact that $x_1\neq y_1$, $x'_1\neq y'_1$ and $x_2\neq x_0$ (see Eq. (18)) we find the six constraints $\mu_1\neq\mu_2$, $\mu_3\neq\mu_4$, $\mu_5\neq\mu_6$, $\mu_7\neq\mu_8$, $\mu_3\neq\mu_8$ and $\mu_4\neq\mu_7$. Nevertheless, evaluation of Eq. (23) remains challenging even under the simplest approximation that we will employ here (though we discuss corrections later on). This approximation assumes that the $V^m_{\mu}$ are independent zero-mean Gaussian random numbers obeying

$$\mathbb{E}\left[V^m_{\mu}\right]=0, \quad \mathbb{E}\left[V^m_{\mu}V^n_{\nu}\right]=\mathbb{E}\left[\bar{V}^m_{\mu}\bar{V}^n_{\nu}\right]=0, \quad \mathbb{E}\left[V^m_{\mu}\bar{V}^n_{\nu}\right]=\delta^{mn}\delta_{\mu\nu}u(m-\mu), \tag{24}$$

with $\delta^{mn}$ and $\delta_{\mu\nu}$ denoting the standard Kronecker symbol with super- or subscripts, respectively. The ensemble average in Eq. (23) can then be evaluated using Isserlis' theorem, which

turns an expectation value of $2n$ random variables into sums over "pairings" where each pairing is a product of $n$ pairs. As an example, consider

$$\mathbb{E}\left[V_3^1 \bar{V}_4^2 V_4^2 \bar{V}_3^1\right] = \mathbb{E}\left[V_3^1 \bar{V}_4^2\right]\mathbb{E}\left[V_4^2 \bar{V}_3^1\right] + \mathbb{E}\left[V_3^1 \bar{V}_3^1\right]\mathbb{E}\left[V_4^2 \bar{V}_4^2\right]$$
$$= \delta^{12}\delta_{34}u^2(m_1 - \mu_3) + u(m_1 - \mu_3)u(m_2 - \mu_4). \tag{25}$$

Quite discomfortingly, the ensemble average in Eq. (23) involves *sixteen* random numbers. In Appendix A.2 a numerical code is detailed that generates all pairings using Isserlis theorem while respecting the ten constraints mentioned above. Then, from the total amount of 40,320 pairings 347 distinct pairings (no multiplicity) survive. It has been found too demanding to write a programme that automatically estimates Eq. (23) and since it is very tiring to investigate 347 cases manually, we look for the most dominant contributions. This is justified because we are only interested in an *order-of-magnitude estimate* of $\mathbb{E}\{[q(x_2, x_0)]^2\}$, not its exact value.

To find the leading order contribution, we observe that each pairing gives rise to a different number of *distinct* Kronecker deltas. In general, the fewer the Kronecker deltas, the larger the contribution because each Kronecker delta "kills" a high dimensional sum. Some care, however, is required because the sums run over spaces with potentially very different dimension. Specifically, every lower subscript runs over a subspace with dimension equal to the rank of some projector $\Pi_x$, which is always smaller than $D$ but could still be comparable to it. In contrast, six out of the eight superscripts run over subspaces with dimension $d \ll D$. Therefore, the leading order contributions are given by the terms that have the fewest Kronecker deltas in total or the fewest Kronecker deltas with respect to the subscripts.

Starting with the latter, the programme from Appendix A.2 shows that the pairing with the fewest amount of subscript-Kronecker deltas is

$$\mathbb{E}\left[V_2^1 \bar{V}_2^4\right]\mathbb{E}\left[V_4^1 \bar{V}_8^6\right]\mathbb{E}\left[V_1^2 \bar{V}_1^3\right]\mathbb{E}\left[V_4^2 \bar{V}_8^5\right]\mathbb{E}\left[V_3^3 \bar{V}_7^8\right]\mathbb{E}\left[V_4^4 \bar{V}_7^7\right]\mathbb{E}\left[V_6^5 \bar{V}_6^8\right]\mathbb{E}\left[V_5^6 \bar{V}_5^7\right]$$
$$\approx D^{-8}\delta^{14}\delta^{16}\delta^{23}\delta^{25}\delta^{38}\delta^{47}\delta_{48}\delta_{37}. \tag{26}$$

Here, all $u(m - \mu) \geq 0$ were replaced for notational simplicity with their maximum value $D^{-1}$ in agreement with the comment below Eq. (14). Then, inserting this term into Eq. (23) and killing all the sums, one obtains the contribution

$$\frac{1}{D_{x_0}^2 D^8}\sum^{1\approx 2}(e^{i\omega^{12}(t_2 - t_1)} - 1)(e^{i\omega^{12}t_1} - 1)(e^{i\omega^{21}(t_2 - t_1)} - 1)(e^{i\omega^{21}t_1} - 1)$$
$$\times \sum_{x_1 \neq y_1}\sum_{x_1' \neq y_1'}\sum_{1,2,3,4,5,6}\chi_1(x_1)\chi_2(y_1)\chi_3(x_0)\chi_4(x_2)\chi_5(x_1')\chi_6(y_1'). \tag{27}$$

Now, since $|e^{i\omega} - 1| \leq 2$ for all $\omega \in \mathbb{R}$ the first line is estimated by setting $e^{i\omega} - 1 = \mathcal{O}(1)$ such that $\sum^{1\approx 2} \mathcal{O}(1) \approx Dd$. The second line can be exactly evaluated by introducing the Hilbert subspace dimension $D_x \equiv \text{tr}\{\Pi_x\} = \sum_1 \chi_1(x)$ associated to the projector $\Pi_x$. Thus, one obtains

$$\frac{1}{D_{x_0}^2 D^8}Dd\sum_{x_1 \neq y_1}\sum_{x_1' \neq y_1'}D_x D_y D_{x_0}D_{x_2}D_{x'}D_{y'} = \frac{dD_{x_2}}{D_{x_0}D^3}\left[1 - \sum_x\left(\frac{D_x}{D}\right)^2\right]^2. \tag{28}$$

We see that even in the worst case scenario, assuming $D_{x_0} \approx 1$ and $D_{x_2} \approx D$, the right hand side scales at least as $d/D^2$, which is certainly negligible small.

We continue by considering the terms with the fewest Kronecker deltas in total. The programme from Appendix A.2 shows that these terms have six Kronecker deltas in total and there are the following four of them (neglecting the universal prefactor $D^{-8}$):

$$\delta^{13}\delta^{57}\delta_{14}\delta_{67}\delta_{23}\delta_{58}, \quad \delta^{13}\delta^{68}\delta_{14}\delta_{68}\delta_{23}\delta_{57}, \quad \delta^{24}\delta^{57}\delta_{24}\delta_{67}\delta_{13}\delta_{58}, \quad \delta^{24}\delta^{68}\delta_{24}\delta_{68}\delta_{13}\delta_{57}. \tag{29}$$

Let us look at the first one. Inserting it into Eq. (23) gives rise to the contribution

$$
\frac{1}{D_{x_0}^2 D^8} \overset{1\approx2\approx4}{\sum} \overset{5\approx6\approx8}{\sum} (e^{i\omega^{12}(t_2-t_1)}-1)(e^{i\omega^{41}t_1}-1)(e^{i\omega^{56}(t_2-t_1)}-1)(e^{i\omega^{85}t_1}-1)
$$
$$
\times \sum_{x_1\neq y_1} \sum_{x_1'\neq y_1'} \sum_{1,2,5,6} \chi_1(x_1)\chi_2(y_1)\chi_2(x_0)\chi_1(x_2)\chi_5(x_1')\chi_6(y_1')\chi_6(x_0)\chi_5(x_2). \tag{30}
$$

Using the same reasoning as above, the summation over the superscripts is approximated as $D^2 d^4$ and the summation over the subscripts gives $\delta_{x_1 x_2}\delta_{y_1 x_0}\delta_{x_1' x_2}\delta_{y_1' x_0}D_{x_0}^2 D_{x_2}^2$. Consequently,

$$
\frac{1}{D_{x_0}^2 D^8} D^2 d^4 D_{x_0}^2 D_{x_2}^2 = \frac{D_{x_2}^2}{D^2}\frac{d^4}{D^4}, \tag{31}
$$

which is still negligible small, though potentially considerably larger than Eq. (28). Using the same strategy, it turns out that also the remaining contributions in Eq. (29) have the same scaling.

Thus, to summarize, it was shown that the dominant contributions to $\mathbb{E}\{[q(x_2, x_0)]^2\}$ scale like $(d/D)^4$, which is negligible small due to the slowness and coarseness of the considered observable $X$. Remarkably, this scaling holds for all times $t_1$ and $t_2$ (and is thus clearly applicable out of equilibrium). Several assumptions concerning the ETH ansatz as detailed in Ref. [15] could be overcome due to the fact that we employed a more transparent but also more restrictive random matrix theory approach from the beginning. Clearly, also the random matrix theory approach is not without assumptions, but recalling its enormous success to deal with non-integrable or chaotic many-body systems [19–24] most assumptions should turn out to be mild in practice.

Nevertheless, a critical point concerns the approximation that the $V_\mu^m$ are Gaussian and uncorrelated. Both cannot be strictly true. First, unitarity implies $|V_\mu^m|^2 \leq 1$, which is not satisfied by a Gaussian distribution. Second, unitarity also implies that $\sum_\mu V_\mu^m \bar{V}_\mu^n = \delta^{mn}$ and $\sum_m V_\mu^m \bar{V}_\nu^m = \delta_{\mu\nu}$, which is satisfied on average, see Eqs. (14) and (24), but not for a single realization if the $V_\mu^m$ are taken uncorrelated. While one might expect corrections to be negligible in many cases due to the huge Hilbert space dimension and the smallness of $V_\mu^m$, Dabelow and Reimann have shown that they can be important [93, 95]. Yet, their goal was to determine the exact time-dependent behaviour of expectation values, instead of the rough order-of-magnitude estimate that we were interested in here. Nevertheless, Appendix A.3 confirms that at least the leading order correction does not give rise to a different scaling. Unfortunately, the author found the calculation of higher order corrections to be intractable, so the present derivation might be best interpreted as strong evidence, but no proof, that slow and coarse observables imply classical measurement statistics in random matrix models and, likely, also beyond.

Finally, we remark that the above derivation made use of the property $x_1 \neq y_1$ in Eq. (18), but the evaluation of the sum $\sum_{x_1\neq y_1}$ has not been crucial. Thus, we have indeed not only shown the Kolmogorov consistency condition but also the strictly stronger decoherent histories condition.

## 3.2 Numerical verification

To illustrate the main features of the present approach to classicality, we use a simple toy model. This model cannot compete with the simulations of more realistic, non-random models in Refs. [7, 8, 15]. Yet, the results clearly support our general findings and they are also used to point out interesting features that were not yet investigated.

The toy model describes energy exchanges between two energy bands and has been studied in detail in Ref. [106]. Each band is described by $N$ equidistant energy levels such that the baseline (unperturbed) Hamiltonian is

$$H_0 = \delta E \sum_{i=0}^{N-1} \frac{i}{N-1} (|i\rangle\langle i| + |i+N\rangle\langle i+N|).  \tag{32}$$

Here, $\delta E$ sets an overall energy scale, which we choose in our numerics equal to $\delta E = 0.5$. Moreover, $|i\rangle$ describes an energy eigenstate of the first (second) band if $i \in \{0,\ldots,N-1\}$ ($i \in \{N,\ldots,2N-1\}$). The random coupling between the bands is mediated by

$$\epsilon H_1 = \epsilon \sum_{i,j=0}^{N-1} v_{ij} |i\rangle\langle j+N| + \text{H.c.},  \tag{33}$$

where the $v_{ij}$ are independent zero mean unit variance Gaussian random numbers. The observable $X$ we consider quantifies the energy imbalance between the two bands and is defined as

$$X = \frac{1}{\sqrt{2N}} \sum_{i=0}^{N-1} (|i+N\rangle\langle i+N| - |i\rangle\langle i|) = \frac{1}{\sqrt{2N}} (\Pi_2 - \Pi_1),  \tag{34}$$

where $\Pi_1$ ($\Pi_2$) is the projector on the first (second) energy band. Clearly, this is a coarse observable with two eigenspaces of dimension $N$. The prefactor $(2N)^{-1/2}$ is pure convention, but bears the advantage that the observable $X$ has the same "size" (meaning that $\text{tr}\{X\} = 0$ and $\text{tr}\{X^2\} = 1$ always) for different $N$. The condition for $X$ to be slow was worked out in Ref. [106] and reads

$$\frac{16\pi^2 N \epsilon^2}{\delta E^2} \ll 1,  \tag{35}$$

i.e., weak coupling gives rise to a slow observable as usual (note that $[H_0, X] = 0$). In the numerical simulations we choose $\epsilon = \epsilon(N)$ such that the left hand side of Eq. (35) becomes 0.01 unless otherwise stated. Moreover, the relaxation time-scale of $X$ is given by $\tau_X = (4\pi\epsilon^2 N)^{-1}\delta E$ [106].

We first consider the structure of the observable $X$ in more detail as it plays an important role in our study. For this purpose Fig. 3 shows matrix elements of the observable $X$ and the projector $\Pi_1$ and we can immediately confirm that $X$ and $\Pi_1$ are narrowly banded. Besides the observation of narrow bandedness, we also note that the off-diagonal elements of $\Pi_1$ appear random and vary erratically. Furthermore, they decay with system size. More specifically, the black circles are roughly one order of magnitude larger than the blue circles, which suggests a scaling $1/\sqrt{D}$ for the off-diagonal elements. These points are in complete agreement with the general predictions of the ETH [23, 24].

Let us now consider the dynamics and test whether the time evolution of $X$ is sensitive to a dephasing operation. To this end, we plot and compare in Fig. 4 the two quantities

$$\sum_{x_s=0}^{1} p(1_t, x_s) = \sum_{x=0}^{1} \text{tr}\left\{\Pi_1 e^{-iHt} \Pi_x e^{-iHs} |\psi_0\rangle\langle\psi_0| e^{iHs} \Pi_x e^{-iHt}\right\},  \tag{36}$$

$$p(1_t, \cancel{x_s}) = \text{tr}\left\{\Pi_1 e^{-iH(t+s)} |\psi_0\rangle\langle\psi_0| e^{iH(t+s)}\right\},  \tag{37}$$

i.e., the probability to find the system in the first energy band at time $t$ with or without dephasing operation at time $s < t$, respectively. Note that Fig. 4 plots these quantities for a single realization of the random matrix Hamiltonian and for a single Haar-randomly chosen initial state confined to the first energy band $\Pi_1$, but (importantly) the results were found to be representative as different realizations of the Hamiltonian or initial state give rise to a

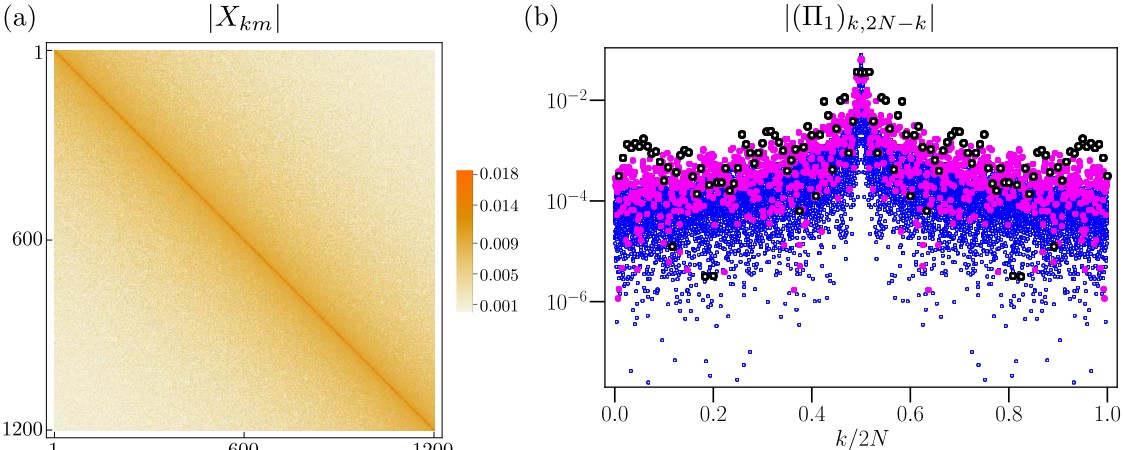

Figure 3: Matrix elements explaining the structure of the considered observable. (a) "Matrix plot" of $|X_{km}|$ in an ordered energy eigenbasis of $H = H_0 + \epsilon H_1$ for $N = 600$ (note that the Hilbert space dimension is $D = 2N$). (b) Plot of the absolute value of the matrix elements of $\Pi_1$ along the "counter-diagonal" (from the lower left to the upper right corner) in an ordered energy eigenbasis for $N = 60$ (larger black circles), $N = 600$ (medium sized pink circles) and $N = 6000$ (tiny blue circles).

similar picture. We also note that the dephasing in Fig. 4 clearly happens *before* the system equilibrates and it becomes immediately evident that the process becomes classical with increasing $N$. This confirms our main result, but Fig. 4 contains two more important pieces of information.

First, the circles and crosses in Fig. 4 are generated for the same Hamiltonian and initial state, but by using *truncated projectors*, which are obtained from $\Pi_x$ by setting all off-diagonal elements with a distance to the diagonal greater than $d/2$ to zero. The choice for $d$ was $d = D^{0.7}$ and the reason for choosing this precise value of the exponent becomes clear later. For now it only matters that we confirm that $d/D = D^{-0.3}$ is very small for large $D$ and that we see in Fig. 4 that even for small $D$ the dynamics is unchanged. This provides numerical evidence that truncating the sums, which was a crucial step to arrive at Eq. (20) in our general derivation, is justified.

Second, another important piece of information is revealed by the number $\Delta$ in the top right corner of each plot. It equals the *trace norm* between the states with and without dephasing defined as

$$\Delta = \Delta(\rho, \mathcal{D}\rho) \equiv \frac{1}{2}\text{tr}\sqrt{(\rho - \mathcal{D}\rho)^2} \in [0, 1], \tag{38}$$

where $\mathcal{D}\rho = \sum_x \Pi_x \rho \Pi_x$ denotes the dephased state. The trace norm is a distance measure characterizing the distinguishability of two quantum states and it has a wide range of applications and favorable properties [104], including that $(1 + \Delta)/2$ is the maximum success probability to distinguish between $\rho$ and $\mathcal{D}\rho$ in an unbiased mixture given *unlimited* measurement power. Thus, $\Delta$ seems to be well suited to measure the amount of coherences in $\rho$. Interestingly, we show in Appendix A.4 that $\max_\rho \Delta(\rho, \mathcal{D}\rho) = 1/2$. Now, observing the values for the trace norm in Fig. 4 we see that they are very close to the maximum possible value. In that sense, the dephasing operation is (almost) maximally invasive and a lot of coherences is destroyed. This demonstrates that (global) decoherence is not needed to explain classical behaviour and *even maximally coherent states can show classical behaviour*. This is not in conflict with OQS decoherence, where decoherence happens locally but usually not globally.

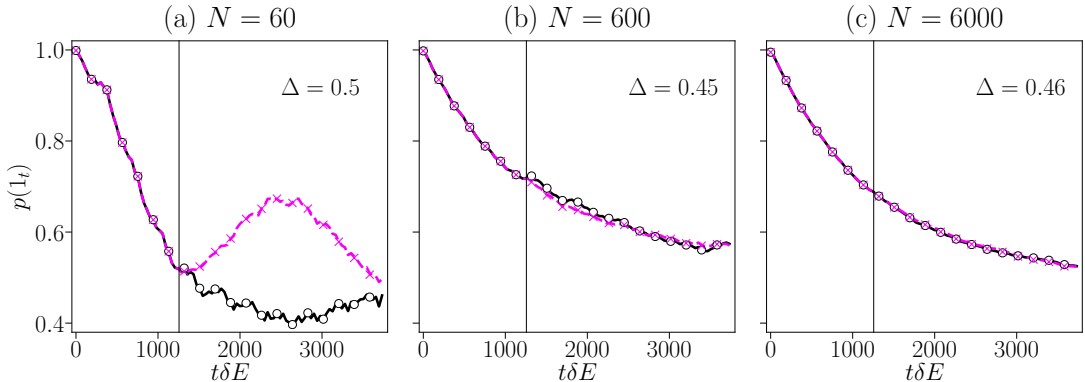

Figure 4: Exemplary check of the Kolmogorov consistency condition for the three system sizes $N = 60$ (a), $N = 600$ (b) and $N = 6000$ (c). We plot Eq. (36) (pink dashed lines) with a dephasing operation at time $s = \tau_X$ (indicated by a black vertical line) and Eq. (37) (black solid lines). The pink crosses and black circles correspond to Eqs. (36) and (37), respectively, but obtained with *truncated* projectors as explained in the main text. Moreover, the value of $\Delta$ in the upper right corner is the trace distance introduced in Eq. (38).

Next, Fig. 5 shows the scaling behaviour of $Q$, Eq. (17), as a function of the Hilbert space dimension. To this end, we plot the time average

$$\langle Q \rangle \equiv \frac{1}{t} \int_s^{t+s} du \left| \sum_{x_s=0}^1 p(1_u, x_s) - p(1_u, \cancel{x_s}) \right|, \tag{39}$$

for $s = \tau_X$ and $t + s = 3\tau_X$, characterizing the (average) distance between the black solid and pink dashed curves in Fig. 4. Moreover, to minimize the risk of statistical outliers, this is done for three different realizations of the random matrix Hamiltonian and three Haar-randomly chosen initial states, thus giving nine realizations for each $N$ as indicated by black circles in Fig. 5. To extract the scaling, we average these nine points for each $N$ and fit a curve of the form

$$\langle Q \rangle \sim \frac{1}{D^\alpha}, \tag{40}$$

which is inspired by Ref. [15]. By looking at Fig. 5 (note the logarithmic scale), one might wonder whether it is a good idea to fit all the data by a straight line (pink dashed line with exponent $\alpha = 0.9$) because the behaviour for $N \lesssim 400$ clearly deviates from the straight line fit obtained for $N \geq 600$ (blue solid line with exponent $\alpha = 0.6$). It is not completely clear to the author what causes the discrepancy, but in Ref. [106] it was observed that the weak coupling approximation requires the side constraint $8\pi^2 N^2 \epsilon^2 / \delta E^2 > 1$, which for our choice of $\epsilon$ implies $N > 200$. This might explain the "anomalous" behaviour for small $N$. Also note that the exponent $\alpha = 0.6$ roughly fits the scaling behaviour observed in Ref. [15] (where $\alpha$ was observed to be in the range $[0.25, 0.6]$).

In any case, we use the fit to determine the number $d$ at which we truncated the projectors to generate the pink crosses and black circles in Fig. 4. Namely, our main result predicted a scaling of the form $(d/D)^4$ for $[q_{t,s}(x_0)]^2$ defined in Eq. (23). This suggests that $\langle Q \rangle$ should scale as $(d/D)^2$. Comparing with $D^{-\alpha}$ for $\alpha = 0.6$ gives $d = D^{0.7}$ as used in Fig. 4.

Finally, we challenge the present approach by relaxing certain assumptions. First, we ask what happens if the initial state is not randomly chosen within the first energy band. Figure 6(a) shows the breakdown of classicality for a highly atypical initial state $|\psi_0\rangle = |i\rangle$ for some randomly selected $i \in \{0, \dots, N-1\}$ for $N = 6000$. Experimentally, preparing such

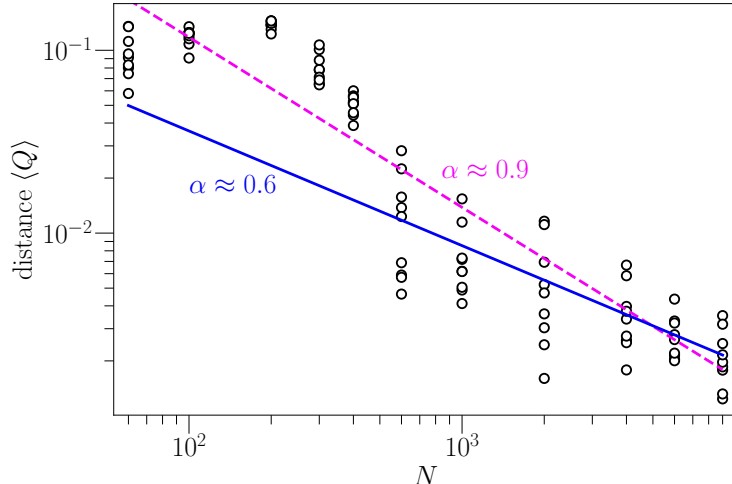

Figure 5: Scaling behaviour of a suitable time average of $Q$ defined in Eq. (39). Black circles correspond to different realizations of the random matrix Hamiltonian or different Haar-randomly chosen initial states confined to the energy band $\Pi_1$ (as in Fig. 4). The lines are obtained from fitting $\langle Q \rangle \sim D^{-\alpha}$ to the average of the black circles including values for all $N$ (pink dashed line) or excluding values for $N \leq 400$ (blue solid line). Note the double logarithmic scale.

an initial state requires precise microscopic control over the eigenstates of each energy band, which clearly violates the agreement made above Eq. (18). Nevertheless, classicality is quickly restored for more realistic states as demonstrated in Fig. 6(b). It shows $\langle Q \rangle$ for $N = 600$, $N = 2000$ and $N = 6000$ for five different realizations of $|\psi_0\rangle = |i\rangle$ (black circles) and for five different initial states $|\psi_0\rangle \sim \sum_{i \in K} c_i |f(i)\rangle$, where $K \subset \{0, \ldots, N-1\}$ is a randomly chosen subset with $0.005N$ many elements (i.e., 0.5% of the energy levels in the first band are initially populated) and the $c_i$ are zero mean unit variance Gaussian random numbers (pink triangles). Despite quite large fluctuations, Fig. 6(b) indicates a scaling law and the pink triangles are (on average) clearly below the black circles, showing the emergence of classicality even for moderately atypical states with a small fraction of populated levels. Finally, Figure 6(c) shows exemplarily what happens for two dephasing operations and an initial random state as used also in Fig. 4. While this is certainly not conclusive, it indicates that for sufficiently large dimensions the here introduced concept of classicality is robust also for $n \geq 3$ measurements.

Last but not least, Fig. 7 investigates the impact of the coupling strength on classicality, which is directly related to the slowness of $X$. Here, weak, medium or strong coupling means that the right hand side of Eq. (35) was fixed to 0.01, 0.1 or 1. One sees that classicality is well satisfied up to medium coupling strength, but fails in the strong coupling regime. This is not a deficit of the present theory because clearly not all observables can behave classical. For strong coupling the eigenenergies of the total Hamiltonian can no longer be approximated by the local eigenenergies of the two bands, but are strongly hybridized, and it is questionable how far $X$ describes any meaningful energy difference in this case.

# 4 Conclusion

The first half of this paper compared and contrasted well established and important approaches to classicality, namely decoherence in OQS and consistent/decoherent histories, with recent abstract research [9–12] as well as numerical evidence [7,8,15] and general derivations [15]

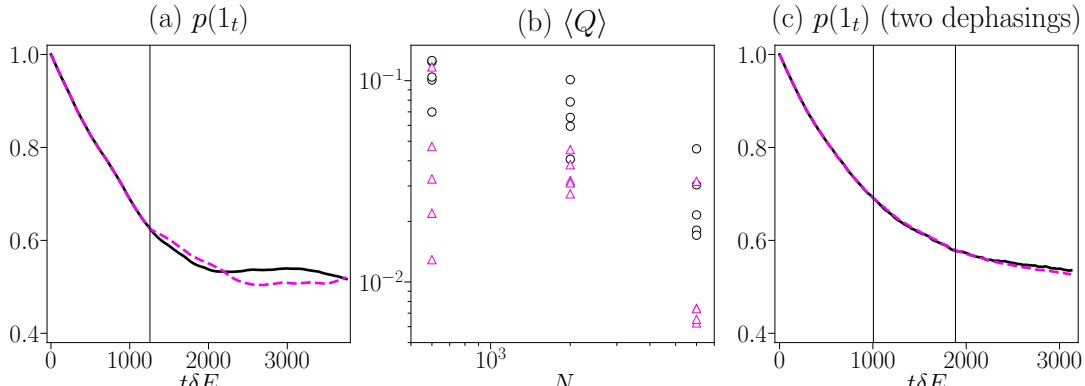

Figure 6: (a) Violation of classicality for a highly atypical state (a random local energy eigenstate $|i\rangle$ in the first energy band) for $N = 6000$ (other parameters and conventions as in Fig. 4). (b) Scaling plot as in Fig. 5 for the just mentioned highly atypical states (black circles) and for random superpositions of $0.005N$ such highly atypical states (pink triangles). (c) Influence of two dephasing operations (indicated by the vertical black lines) for $N = 6000$ and a typical nonequilibrium initial state as considered in Fig. 4.

of classicality based on the Kolmogorov consistency condition. Arguably, the difference between the consistent/decoherent histories condition and the Kolmogorov consistency condition is small. However, Kolmogorov consistency is easier to verify experimentally than consistent/decoherent histories and it can be independently well motivated from an operational perspective. Moreover, we established that quantum Markovianity is a key concept to relate decoherence in OQS to both the consistent/decoherent histories and the Kolmogorov consistency condition. Figure 2 summarizes the first part.

The second half of the paper has given for an experimentally relevant class of initial states an independent derivation of the Kolmogorov consistency and the decoherent histories condition based on a random matrix theory model. We carefully checked numerically the correctness of the involved approximations. Remarkably, it was explicitly shown that even maximally coherent states can give rise to classical dynamics for global observables.

Several interesting research avenues open up for the future. For instance, classicality could here be only established for "mini-histories" with two measurement results and extending the derivation to longer histories, as done in a different context in Refs. [100–103], is highly desirable. Indeed, recent numerical results for up to five-time histories have confirmed that the emergence of classicality is a robust phenomenon [107]. Moreover, various fundamental question might appear in a new light, for instance, the relationship between quantum Darwinism [36–38] and quantum Markovianity, applications of decoherent histories to OQS theory [108, 109], or the implications of the present findings for quantum cosmology [35]. Finally, the central quantity investigated in Eq. (18) can be more generally seen as a particular example of a Kirkwood-Dirac quasiprobability [110]. The behaviour of these quasiprobabilities for chaotic systems has also raised attention in relation to out-of-time-ordered correlators [111–113], which might open up interesting possibilities for fruitful connections between different fields.

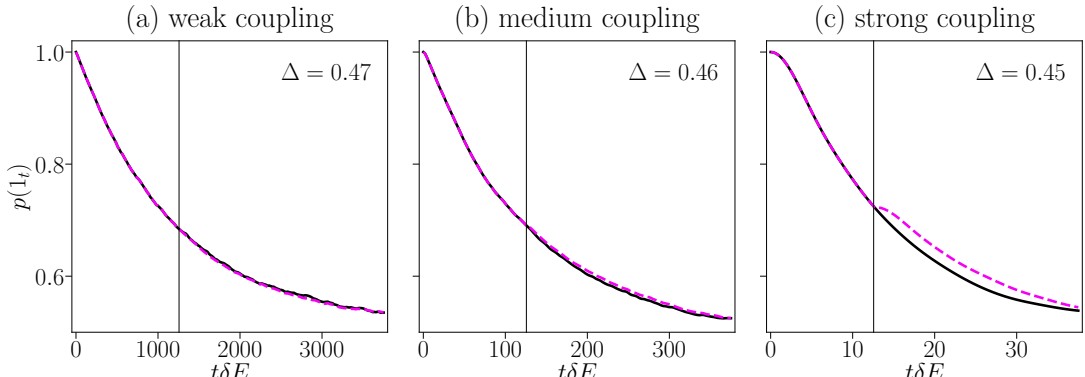

Figure 7: Exemplary violation of classicality at strong coupling/for fast $X$. Plots are done as in Fig. 4 for $N = 6000$ and by choosing $\epsilon$ such that the right hand side of Eq. (35) equals 0.01 (weak coupling, as before (a)), 0.1 (medium coupling (b)) and 1 (strong coupling (c)). Note that the relaxation time scale changes inversely proportional to it.

# Acknowledgements

This manuscript has significantly benefited from discussions with and feedback from Lennart Dabelow about random matrix theory, Wojciech Zurek about the quantum-to-classical transition, and John Calsamiglia and Andreas Winter about trace distance bounds. For further discussions and feedback I thank Victor Bastidas, Giulio Gasbarri, Jochen Gemmer, Nicole Yunger Halpern, Joseph Schindler and Jiaozi Wang.

**Funding information** The author is finanically supported by "la Caixa" Foundation (ID 100010434) under the fellowship code LCF/BQ/PR21/11840014 and co-funded by the Spanish Agencia Estatal de Investigación (project no. PID2019-107609GB-I00), the Spanish MINECO (FIS2016-80681-P, AEI/FEDER, UE), the Generalitat de Catalunya (CIRIT 2017-SGR-1127), and the European Commission QuantERA grant ExTRaQT (Spanish MICINN project PCI2022-132965).

# A Appendix

## A.1 Decoherence, Markovianity and consistency

This appendix assumes the reader to be familiar with superoperators, in particular instruments and completely positive (and trace preserving) maps. Introductions are provided, e.g., in Refs. [13, 14, 43, 104]. All superoperators are denoted with calligraphic symbols $\mathcal{A}, \mathcal{E}, \mathcal{P}, \dots$.

We start by defining a quantum Markov process following Ref. [16], for introductory treatments see Refs. [13, 14]. This definition recognizes the crucial role played by an external agent (or experimenter or observer), who interrogates or intervenes the dynamics of an OQS at a set of discrete times $\{t_n, \dots, t_2, t_1\}$. Each intervention at each time $t_k$ is described by an instrument $\{\mathcal{A}_k(r_k)\}$, which is a set of completely positive maps $\mathcal{A}_k(r_k)$ adding up to a completely positive and trace preserving map $\mathcal{A}_k \equiv \sum_{r_k} \mathcal{A}_k(r_k)$. Importantly, these maps only act on the system Hilbert space, encapsulating the idea that the external agent has no control over the bath degrees of freedom. Moreover, $r_k$ denotes some abstract measurement outcome, not necessarily related to the $x_k$ appearing in the main text. Now, a quantum process is Markovian if the re-

sponse of the OQS to any sequence (or history) of interventions $\{\mathcal{A}_n(r_n), \ldots, \mathcal{A}_2(r_2), \mathcal{A}_1(r_1)\}$ can be written as

$$\tilde{\rho}_S(t_n | r_n, \ldots, r_2, r_1) = \mathcal{A}_n(r_n)\mathcal{E}_{n,n-1}\cdots\mathcal{A}_2(r_2)\mathcal{E}_{2,1}\mathcal{A}_1(r_1)\mathcal{E}_{1,0}\rho_S(t_0), \qquad (A.41)$$

where $\{\mathcal{E}_{k,k-1}\}_{k=1}^n$ is a set of completely positive and trace preserving maps, which—importantly—do not depend on the interventions or initial system state. Moreover, $\tilde{\rho}_S(t_n | r_n, \ldots, r_2, r_1)$ is the subnormalized OQS state conditioned on the sequence of interventions, which happens with probability $p(r_n, \ldots, r_2, r_1) = \mathrm{tr}_S\{\tilde{\rho}_S(t_n | r_n, \ldots, r_2, r_1)\}$. Finally, notice that the $\mathcal{E}_{k,k-1}$ are also known as *dynamical maps* as they prograpate the system state forward in time from $t_{k-1}$ to $t_k$. These maps encode the influence of the bath or environment and, according to Eq. (A.41), a Markov process is precisely characterized by the fact that the influence of the bath can be neatly separated from the interventions $\mathcal{A}_k(r_k)$ of the external agent.

A few more words of clarification might be helpful. First, one can show that Eq. (A.41) reduces to the classical Markov condition in an appropriate limit. Second, for classical causal models it reduces to the causal Markov condition of Ref. [42]. Third, the validity of Eq. (A.41) can be checked by local interventions on the system only. Fourth, the existence of a Markovian quantum master equation for $\rho_S(t)$, as often studied in OQS theory, does *not* imply the validity of Eq. (A.41), although the converse is true (see in particular Ref. [18]). Note that the identity operator $\mathcal{I}$ ("do nothing!") is an instrument too such that it is meaningful to define the dynamical map $\mathcal{E}_{\ell,k} \equiv \mathcal{E}_{\ell,\ell-1}\cdots\mathcal{E}_{k+1,k}$ for any $\ell - k > 1$. Fifth, Eq. (A.41) really is a statement about the multi-time behaviour of an OQS, and the validity of Eq. (A.41) can be shown to be equivalent to an appropriate formulation of the quantum regression theorem [114].

Next, we give a rigorous mathematical definition of OQS decoherence and a derivation that it implies the decoherent histories condition for quantum Markov processes (which in turn implies Kolmogorov consistency). To the best of the author's knowledge, no definition of OQS decoherence exists and the notion is used rather conceptually (what follows is, however, closely related but not identical to the treatment of Refs. [9, 10, 12]). However, if one wants to prove things mathematically, one has to start with a definition. For this purpose, we use the dephasing operation $\mathcal{D}$ in the pointer basis as introduced in Sec. 2.2. Then, for a quantum Markov process we define OQS decoherence by requiring that

$$[\mathcal{E}_{\ell,k}, \mathcal{D}] = 0, \quad \text{for all} \quad n \geq \ell > k \geq 1, \qquad (A.42)$$

where $[\mathcal{A}, \mathcal{B}] = \mathcal{A}\mathcal{B} - \mathcal{B}\mathcal{A}$ is the commutator in superoperator space.

Is this a good definition of OQS decoherence? At least it implies that the dynamics induced by the environment is not able to create coherences in the pointer basis. To see this, we introduce the superoperator $\mathcal{P}_{x,y}\rho \equiv \Pi_x^S \rho \Pi_y^S$. Next, suppose that $\rho_S = \mathcal{D}\rho_S$ is some system state without coherences. Then, Eq. (A.42) implies that $\mathcal{P}_{x,y}\mathcal{E}\rho_S = 0$ for all $x \neq y$, i.e., it is not possible to create coherences in the pointer basis when starting from a decohered state. Clearly, this captures a key aspect of the OQS decoherence concept, but one could naturally impose further constraints. For instance, Eq. (A.42) makes no statement about the decoherence time $t_{\mathrm{dec}}$ and, since the short time dynamics of OQS is complex, one might additionally require that Eq. (A.42) is only valid on a coarse time scale, i.e., for $t_\ell - t_k$ not too small. In any case, the minimal definition given here turns out to be sufficient to prove that histories in the sense of Eq. (8) are decoherent.

To see this, we conveniently write the decoherence functional for a quantum Markov process using superoperators:

$$\mathfrak{D}(\mathbf{x}; \mathbf{y}) = \mathrm{tr}_S\left\{\mathcal{P}_{x_n,y_n}\mathcal{E}_{n,n-1}\mathcal{P}_{x_{n-1},y_{n-1}}\mathcal{E}_{n-1,n-2}\cdots\mathcal{P}_{x_1,y_1}\mathcal{E}_{1,0}\rho_S(t_0)\right\}. \qquad (A.43)$$

Next, note that the decoherence functional does not change when subjecting it to a final dephasing operation $\mathcal{D}$ in the pointer basis (in fact, the decoherence functional does not change under any final dephasing):

$$\mathfrak{D}(\mathbf{x};\mathbf{y}) = \operatorname{tr}_S \left\{ \mathcal{D}\mathcal{P}_{x_n,y_n}\mathcal{E}_{n,n-1}\mathcal{P}_{x_{n-1},y_{n-1}}\mathcal{E}_{n-1,n-2}\cdots\mathcal{P}_{x_1,y_1}\mathcal{E}_{1,0}\rho_S(t_0) \right\}. \tag{A.44}$$

Now, let $k$ be the first index for which $x_k \neq y_k$, i.e., $x_\ell = y_\ell$ for all $\ell > k$. By the definition of OQS decoherence, we can then permute $\mathcal{D}$ through until we hit the time $t_k$:

$$\mathfrak{D}(\mathbf{x};\mathbf{y}) = \operatorname{tr}_S \left\{ \mathcal{P}_{x_n,y_n}\mathcal{E}_{n,n-1}\cdots\mathcal{E}_{k+1,k}\mathcal{D}\mathcal{P}_{x_k,y_k}\mathcal{E}_{k,k-1}\cdots\mathcal{P}_{x_1,y_1}\mathcal{E}_{1,0}\rho_S(t_0) \right\}. \tag{A.45}$$

Finally, elementary algebra shows that $\mathcal{D}\mathcal{P}_{x_k,y_k}\rho = 0$ whatever the input state $\rho$ is. QED.

It is interesting to note that strictly weaker conditions suffice to show Kolmogorov consistency for quantum Markov processes [9, 10, 12], but they seem insufficient to show the decoherent histories condition.

## A.2 Numerical implementation

This appendix includes some details about how to numerically fascilitate the evaluation of the expectation values appearing in Eq. (23) using Mathematica [115].

We are interested in expectation values of the form $\mathbb{E}[V_{\mu_4}^{m_1}\bar{V}_{\mu_4}^{m_2}V_{\mu_1}^{m_2}\bar{V}_{\mu_1}^{m_3}\dots]$ with an equal amount of $V$- and complex conjugate $\bar{V}$-terms. Since each pair in Isserlis theorem requires one $V$- and one $\bar{V}$-term, not all permutations of $(V_{\mu_4}^{m_1},\bar{V}_{\mu_4}^{m_2},V_{\mu_1}^{m_2},\bar{V}_{\mu_1}^{m_3},\dots)$ contribute to the expectation value. One way to create all contributing permutations consists in generating two lists $A = \{\{m_1,\mu_4\},\{m_2,\mu_1\},\dots\}$ and $B = \{\{m_2,\mu_4\},\{m_3,\mu_1\},\dots\}$ associated to the $V$- and $\bar{V}$-terms, respectively, followed by

```
PermA = Permutations[A];
Pairings = Map[Sort, Table[Flatten[PermA[[α, k]], B[[k]]], {α, 1, L_A!}, {k, 1, L_A}], 2];
```

Here, $L_A = \text{Length}[A]$ denotes the lengths of the list $A$ (which equals $L_B$) and consequently $L_A!$ is the length of $PermA$. The output $Pairings$ now contains all possible pairings of the form

$$\begin{aligned} Pairings = \{&\{\{m_1,m_2,\mu_4,\mu_4\},\{m_2,m_3,\mu_1,\mu_1\},\dots\}, \\ &\{\{m_2,m_2,\mu_1,\mu_4\},\{m_1,m_3,\mu_4,\mu_1\},\dots\},\dots\}. \end{aligned} \tag{A.46}$$

The lowest level angular bracket $\{\dots\}$ contains one specific pair with always two Latin and two Greek indices. The middle level angular bracket contains the product of all pairs, which form one specific "pairing".

In general, $Pairings$ will contain many forbidden pairings due to constraints such as $m_1 \neq m_2$, $\mu_1 \neq \mu_2$, etc. To filter them out, we map each pairing to a graph with vertices $(m_1,m_2,\dots,\mu_1,\mu_2,\dots)$ and edges created by the Kronecker symbols of each pair, e.g., the pair $\{m_1,m_2,\mu_4,\mu_4\}$ creates an edge between $m_1$ and $m_2$ and a (redundant) edge between $\mu_4$ and $\mu_4$. Then, e.g., to respect the constraint $m_1 \neq m_2$, a pairing is only accepted if there exists no path in the graph from $m_1$ to $m_2$, see Fig. 8 for a sketch. As an example, the following code creates a list $accepted$ that stores the numbers $i$ for which the $i$th element of $Pairings$

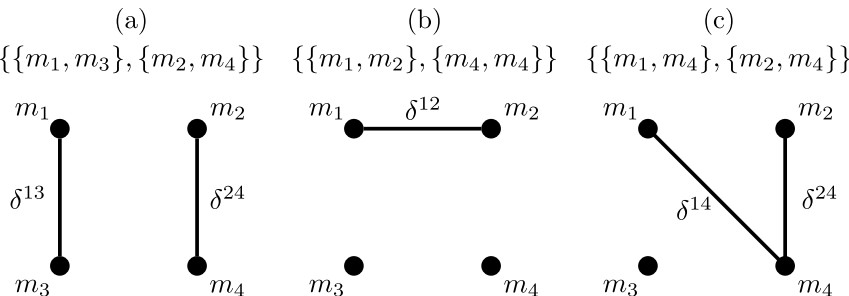

Figure 8: Three example pairings (here, each pairing has two pairs with Latin indices) and the associated graphs. The constraint $m_1 \neq m_2$ is violated in example (b) and (c).

satisfies the constraints $m_1 \neq m_2$ (further constraints can be easily included):

```
For[i = 1, i ≤ L_A!, i++,
 network = {};
 For[j = 1, j ≤ L_A, j++,
  network = Append[network, Pairings[[i,j]][[1]] •–• Pairings[[i,j]][[2]]];
 ];
 G = Graph[network];
 test = Length[Flatten[FindPath[G, m1, m2]]];
 If[test == 0, accepted = Append[accepted, i]];
]
```

Here, the graph $G$ for each pairing $i$ is created using a set of edges stored in $network$, where each edge is symbolized by •–• (typeset as "Esc ue Esc" in Mathematica).

Next, we create a list $Rem$ that simply contains all remaining pairings that satisfy the constraints above. This can be done by

$$Rem = \text{Map}[\text{Sort}, \text{Table}[Pairings[[accepted[[k]]]], \{k, 1, L_{accepted}\}], 2]; \tag{A.47}$$

Note that Sort brings the elements of the pairing in a standard form again.

As hinted at already above, the structure of the pairings can be nicely illustrated with a graph with edges indicating Kronecker deltas. For further manipulation, we now like to convert Rem into a list of graphs:

```
graphs = {};
vert = {m1, m2, …, μ1, μ2, …};
For[i = 1, i ≤ L_accepted, i++,
 medges = Table[Rem[[i,j]][[1]] •–• Rem[[i,j]][[2]], {j, 1, Length[Rem[[1]]]}];
 μedges = Table[Rem[[i,j]][[3]] •–• Rem[[i,j]][[4]], {j, 1, Length[Rem[[1]]]}];
 edges = Join[medges, μedges];
 graphs = Append[graphs, Graph[vert, edges]];
]
con = Map[Sort]@*ConnectedComponents/@graphs;
```

The final output $con$ contains the connectivity of each graph associated to each accepted pairing. For instance, if $\{\{m_1, m_1, \mu_1, \mu_3\}, \{m_2, m_3, \mu_2, \mu_4\}, \{m_2, m_4, \mu_4, \mu_4\}\}$ is one accepted pair-

ing, then its connectivity is $\{\{m_2, m_3, m_4\}, \{\mu_1, \mu_3\}, \{\mu_2, \mu_4\}, \{m_1\}\}$. This format has nice properties as it directly reveals the "structure" of each pairing in terms of (multi-valued) Kronecker deltas. To find out whether there are multiple pairings with the same structure, one can run $\texttt{Tally}[con]$.

Given the connectivity list $con$, it is also straightforward to count the total number of sums that get killed due to Kronecker deltas. In the following, the function $kills$ computes this number for a given pairing and $final$ stores these numbers for each element of $con$:

$$kills[list\_] := \texttt{Total}[\texttt{Map}[\texttt{Length}, list]] - \texttt{Length}[list];$$
$$final = \texttt{Table}[kills[con[[m]]], \{m, 1, L_{accepted}\}];$$

Clearly, the above procedure can be also applied to the graphs formed by Greek indices only—as we needed to do to find the terms with the fewest amount of subscript-Kronecker deltas around Eq. (26).

## A.3 Higher order corrections

In Ref. [93] (see also Sec. 3.4.4 of Ref. [105]) Dabelow and Reimann developed a systematic way to take into account correlations among matrix elements, which we briefly summarize here. Unfortunately, it will turn out that this procedure becomes quickly untractable due to the fact that we already start with an expectation value over a product of sixteen random numbers. Moreover, the method does not take into account correlations with respect to both the perturbed and unperturbed bases $|m\rangle$ and $|\mu\rangle$, but only with respect to one of them. Therefore, while that method was found to work well in Refs. [93, 95, 105], it still does not provide an exact treatment of the problem.

To start with, notice that the matrix element $V_\mu^m$ can be seen as the $m$'th component of a $D$-dimensional vector $V_\mu$. To each $V_\mu$ we associate two vectors $v_\mu$ and $w_\mu$, where the components of $v_\mu$ are assumed to be independent Gaussian random variables with statistical properties equal to those of Eq. (24). Then, what was effectively done in the main text was to approximate

$$\mathbb{E}\left[V_4^1 \bar{V}_4^2 V_1^2 \bar{V}_1^3 V_3^3 \bar{V}_3^4 V_2^4 \bar{V}_2^1 V_8^5 \bar{V}_8^6 V_5^6 \bar{V}_5^7 V_7^7 \bar{V}_7^8 V_6^8 \bar{V}_6^5\right]$$
$$\approx \mathbb{E}\left[v_4^1 \bar{v}_4^2 v_1^2 \bar{v}_1^3 v_3^3 \bar{v}_3^4 v_2^4 \bar{v}_2^1 v_8^5 \bar{v}_8^6 v_5^6 \bar{v}_5^7 v_7^7 \bar{v}_7^8 v_6^8 \bar{v}_6^5\right], \tag{A.48}$$

which disregards all correlations. Instead, we now replace

$$\mathbb{E}\left[V_4^1 \bar{V}_4^2 V_1^2 \bar{V}_1^3 V_3^3 \bar{V}_3^4 V_2^4 \bar{V}_2^1 V_8^5 \bar{V}_8^6 V_5^6 \bar{V}_5^7 V_7^7 \bar{V}_7^8 V_6^8 \bar{V}_6^5\right]$$
$$\approx \mathbb{E}\left[w_4^1 \bar{w}_4^2 w_1^2 \bar{w}_1^3 w_3^3 \bar{w}_3^4 w_2^4 \bar{w}_2^1 w_8^5 \bar{w}_8^6 w_5^6 \bar{w}_5^7 w_7^7 \bar{w}_7^8 w_6^8 \bar{w}_6^5\right], \tag{A.49}$$

and obtain the vectors $\{w_\mu\}$ from $\{v_\mu\}$ using a Gram-Schmidt procedure, which orthonormalizes the set $\{v_\mu\}$, thereby taking into account constraints imposed by the unitarity of $V_\mu^m$. Thus, starting from $w_1 = v_1$, we set[5]

$$w_\mu = v_\mu - \sum_{\nu=1}^{\mu-1} \langle w_\nu | v_\mu \rangle w_\nu, \tag{A.50}$$

for all $\mu \geq 2$ and where $\langle w | v \rangle$ denotes the standard complex scalar product. Inserting the $w_\mu$ in Eq. (A.49) gives an explicit expression in terms of the independent Gaussian variables $v_\mu^m$ that can be calculated using Isserlis' theorem and takes into account correlations.

---

[5]To be precise, the here presented procedure only ensures orthogonality, but not normalization. However, since the real and imaginary coefficients of $v_\mu$ are each drawn from a zero mean (approximate) Gaussian distribution with variance $1/2D$, it follows that $v_\mu$ is not only normalized on average, but each single realization of $v_\mu$ is strongly concentrated around vectors with unit norm for large $D$ [116].

Unfortunately, we would need to do this for eight vectors in Eq. (A.49) (and their complex conjugates) and the total number of $v_\mu^m$-terms, and consequently the number of pairings in Isserlis' theorem, quickly grows to astronomically large numbers, even when respecting the constraints identified in the main text ($m_1 \neq m_2$, $\mu_1 \neq \mu_2$, etc.). To see this, it might be helpful to explicity write down the components obtained via the Gram-Schmidt procedure. Clearly, everything is simple for the first vector: $w_1^m = v_1^m$. The second vector is also still managable: $w_2^m = v_2^m - \sum_n \bar{v}_1^n v_2^n v_1^m$. The third vector, however, contains already six terms with up to seven $v$-components:

$$
\begin{aligned}
w_3^m = v_3^m &- \sum_n \bar{v}_1^n v_3^n v_1^m - \sum_n \bar{v}_2^n v_3^n v_2^m + \sum_{no} v_1^o \bar{v}_2^o \bar{v}_1^n v_3^n v_2^m + \sum_{np} \bar{v}_2^n v_3^n \bar{v}_1^p v_2^p v_1^m \\
&- \sum_{nop} v_1^o \bar{v}_2^o \bar{v}_1^n v_3^n \bar{v}_1^p v_2^p v_1^m ,
\end{aligned}
\tag{A.51}
$$

and it does not get simpler for the remaining vectors.

However, recall that we are only interested in an order-of-magnitude estimate. Each added pair of $v$-terms comes with an extra $D$-dimensional summation, but also contributes a factor of the order $D^{-1}$ due to Eq. (24). In general, one therefore expects that these contributions roughly cancel each other in an order-of-magnitude estimate provided that the minimum number of Kronecker deltas as identified in the main text remains the same (if one term in Isserlis' theorem gives rise to fewer Kronecker deltas than before, then an additional sum appears potentially contributing a huge factor).

Whether this is the case has been explicitly checked to lowest order in the Gram-Schmidt procedure. This means one first sets

$$
w_\mu \approx v_\mu - \sum_{\nu=1}^{\mu-1} \langle v_\nu | v_\mu \rangle v_\nu ,
\tag{A.52}
$$

for $\mu \in \{2, 3, \ldots, 8\}$, which follows from Eq. (A.50) by replacing $w_\mu$ by $v_\mu$ on the right hand side. This approximation is then inserted into Eq. (A.49) and only terms with a single additional sum are kept. There are $2 \cdot (1 + 2 + \cdots + 7) = 56$ such single sum contributions, where the factor two arises because one has to take into account $w_\mu$ and $\bar{w}_\mu$ for $\mu \in \{2, 3, \ldots, 8\}$. However, from the structure of the problem it does not appear that the complex conjugate entries contribute differently (since we are interested in a real-valued object), so these additional terms can be neglected. On the other hand, which of the 28 terms gives the worst contribution to the scaling is not clear.

Therefore, this has been tested with the programme from Appendix A.2 and it was found that each correction term contains enough (multi-valued) Kronecker deltas to kill at least 6 sums, i.e., the same number as identified in Eq. (29). The following table explicitly lists the 28 contributions together with their "Kronecker order" $L(\delta)$, which equals the number of sums killed or, equivalently, the minimum of the list $final$ defined at the end of Sec. A.2.

| replace... | by... | $L(\delta)$ | replace... | by... | $L(\delta)$ |
|---|---|---|---|---|---|
| $w_4^1$ | $-\sum^9 \bar{v}_1^9 v_4^9 v_1^1$ | 6 | $w_5^6$ | $-\sum^9 \bar{v}_2^9 v_5^9 v_2^6$ | 7 |
| | $-\sum^9 \bar{v}_2^9 v_4^9 v_2^1$ | 6 | | $-\sum^9 \bar{v}_3^9 v_5^9 v_3^6$ | 7 |
| | $-\sum^9 \bar{v}_3^9 v_4^9 v_3^1$ | 6 | | $-\sum^9 \bar{v}_4^9 v_5^9 v_4^6$ | 7 |
| $w_3^3$ | $-\sum^9 \bar{v}_1^9 v_3^9 v_1^3$ | 6 | $w_7^7$ | $-\sum^9 \bar{v}_1^9 v_7^9 v_1^7$ | 7 |
| | $-\sum^9 \bar{v}_2^9 v_3^9 v_2^3$ | 6 | | $-\sum^9 \bar{v}_2^9 v_7^9 v_2^7$ | 7 |
| $w_2^4$ | $-\sum^9 \bar{v}_1^9 v_2^9 v_1^4$ | 6 | | $-\sum^9 \bar{v}_3^9 v_7^9 v_3^7$ | 7 |
| $w_8^5$ | $-\sum^9 \bar{v}_1^9 v_8^9 v_1^5$ | 7 | | $-\sum^9 \bar{v}_4^9 v_7^9 v_4^7$ | 7 |
| | $-\sum^9 \bar{v}_2^9 v_8^9 v_2^5$ | 7 | | $-\sum^9 \bar{v}_5^9 v_7^9 v_5^7$ | 6 |
| | $-\sum^9 \bar{v}_3^9 v_8^9 v_3^5$ | 7 | | $-\sum^9 \bar{v}_6^9 v_7^9 v_6^7$ | 6 |
| | $-\sum^9 \bar{v}_4^9 v_8^9 v_4^5$ | 7 | $w_6^8$ | $-\sum^9 \bar{v}_1^9 v_6^9 v_1^8$ | 7 |
| | $-\sum^9 \bar{v}_5^9 v_8^9 v_5^5$ | 6 | | $-\sum^9 \bar{v}_2^9 v_6^9 v_2^8$ | 7 |
| | $-\sum^9 \bar{v}_6^9 v_8^9 v_6^5$ | 6 | | $-\sum^9 \bar{v}_3^9 v_6^9 v_3^8$ | 7 |
| | $-\sum^9 \bar{v}_7^9 v_8^9 v_7^5$ | 6 | | $-\sum^9 \bar{v}_4^9 v_6^9 v_4^8$ | 7 |
| $w_5^6$ | $-\sum^9 \bar{v}_1^9 v_5^9 v_1^6$ | 7 | | $-\sum^9 \bar{v}_5^9 v_6^9 v_5^8$ | 6 |

Finally, one might worry that, even when each single contribution to the expectation value is very small, the sum of the enormous amount of terms involved gives rise to a giant prefactor. However, this is unlikely a problem because, first, this prefactor does not scale with the particle number $N$ and, second, recall that the contributions have different signs, see, e.g., Eq. (A.51). Additonal cancellations are therefore likely and were indeed an important observation in Refs. [93,95].

## A.4 Trace distance bound under dephasing

The author owes the details of the following proof to Ref. [117].

We start by noting that strong convexity of the trace norm [104] implies

$$\max_\rho \Delta(\rho, \mathcal{D}\rho) = \max_\psi \Delta(\psi, \mathcal{D}\psi), \tag{A.53}$$

where $\psi = |\psi\rangle\langle\psi|$ is here used to denote pure states. Next, any such $|\psi\rangle$ can be written as $|\psi\rangle = \sum_{x=1}^M \alpha_x |\psi_x\rangle$ with $\Pi_y |\psi_x\rangle = \delta_{x,y} |\psi_x\rangle$ and $|\alpha_x|^2 = \langle\psi|\Pi_x|\psi\rangle$ such that $\sum_x |\alpha_x|^2 = 1$. One then finds

$$\Delta(\psi, \mathcal{D}\psi) = \frac{1}{2}\text{tr}\sqrt{\left(\sum_{x,y=1}^M \alpha_x \alpha_y^* |\psi_x\rangle\langle\psi_y| - \sum_{x=1}^M |\alpha_x|^2 |\psi_x\rangle\langle\psi_x|\right)}. \tag{A.54}$$

Since the $\{|\psi_x\rangle\}$ are orthonormal for different $x$ and because the trace norm is invariant under unitary rotations [104] such that we can map $|\psi_x\rangle$ to some fixed standard vector for each subspace $x$, Eq. (A.54) makes it evident that we can restrict the problem to an $M$-dimensional Hilbert space, i.e.,

$$\max_\rho \Delta(\rho, \mathcal{D}\rho) = \max_{\tilde{\psi}} \Delta\left(\tilde{\psi}, \tilde{\mathcal{D}}\tilde{\psi}\right), \tag{A.55}$$

where $|\tilde{\psi}\rangle \in \mathbb{C}^M$ and the dephasing operation becomes $\tilde{\mathcal{D}}\tilde{\rho} = \sum_{x=1}^M |x\rangle\langle x|\tilde{\rho}|x\rangle\langle x|$ for some set of one-dimensional projectors $\{|x\rangle\langle x|\}$ spanning $\mathbb{C}^M$.

Next, we note that we can write the dephasing map as $\tilde{\mathcal{D}}\tilde{\rho} = \frac{1}{M}\sum_{k=0}^{M-1} Z^k \tilde{\rho} Z^{-k}$ with the $M$-dimensional diagonal phase unitary $Z = \sum_{x=1}^{M} e^{2\pi i(x-1)/M}|x\rangle\langle x|$. Thus, $\tilde{\rho} - \tilde{\mathcal{D}}\tilde{\rho} = \frac{1}{M}\sum_{k=1}^{M-1}(\tilde{\rho} - Z^k \tilde{\rho} Z^{-k})$ and it then follows from the triangle inequality that

$$\Delta(\tilde{\psi}, \tilde{\mathcal{D}}\tilde{\psi}) \leq \frac{1}{M}\sum_{k=1}^{M-1} \Delta\left(\tilde{\psi}, Z^k \tilde{\psi} Z^{-k}\right) \leq \frac{M-1}{M}. \tag{A.56}$$

Finally, we confirm that the upper bound is satisfied by the maximally coherent state $|\tilde{\psi}\rangle = \sum_x |x\rangle/\sqrt{M}$. Thus, for $M = 2$ we obtain the result used in the main text.

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
