# Peer review of "Classicality with(out) decoherence: Concepts, relation to Markovianity, and a random matrix theory approach"

_SciPost Physics, doi:SciPost Phys. 15, 024 (2023)_

## Round 1 · Referee Report · Anonymous (Referee 1) · 2023-2-28

Report

This paper is a study of the emergence of classicality in Hamiltonian quantum systems through the behaviour of coarse-grained observables. The author first analyses a number of different concepts in the literature, and derives relations between them. Then he derives the conditions under which his favoured concept of classicality, the validity of the Kolmogorov condition for observations, can be derived. This is done by considering when a classical-like observables can be found that is suitably coarse-grained and slow, for a Hamiltonian drawn at random according to random matrix theory. Next, he studies this numerically, in detail, for quite large systems, and finds quite good agreement with his theory.

The paper is generally written with great care and honesty, which is appreciated. In most parts, it certainly builds a lot on work by previous authors. But it seems to me that it contains substantial conceptual, analytical, and numerical advances, and so deserves publication in SciPost, once the comments below are addressed.

Requested changes

  1. “Emergence of classicality” could mean many things to many people. In the introduction, one pair of approaches that the author mentions because it is going to be relevant in the paper (as indeed it is) is decoherent histories / consistent histories. In my understanding these were proposed as an interpretation of quantum mechanics, a solution to the quantum measurement problem. By contrast, the viewpoint adopted by the author in this paper seems very much an operational one: to study “an experimentally well-defined procedure to access the time evolution of a quantum system”. That is, he assumes the existence of measurements as a primitive, and thus is not addressing the measurement problem at all. That is fine in itself, but it would help the reader greatly to make this clear very early in the paper. That is, the author should make clear the scope of answer he is considering to the question he raises in paragraph 1, “what needs to be proved?” Or, if this is not what the author meant by that question, then it seems that there is an even more basic question that the author needs to ask, and answer for the purpose of this paper.

  2. For the relations in Fig. 2, are all the (non-double-headed) arrows one-way? That is, in all cases do there exist counter-examples to show that the concepts are not equivalent? If so, then the author should spell-out (where not already done) the required counter-examples. If the author cannot find a counter-example, then it would be useful to introduce a distinction between arrows known to be one-way, and arrows which could possibly be two-way.

  3. In the first sentence of the second-last paragraph on page 7, it would help to be explicit that what is sufficient is Markovianity on top of Eq.(3). Assuming that my understanding is correct.

  4. In the first paragraph on page 12, dim H_S << dim H_B may be a sufficient condition for coarseness. But I don’t see that it follows from the previously discussed sufficient condition, which would translate to merely requiring 1 << dim H_B.

  5. In the final paragraph on Sec. 2, why is it “unfortunate” that certain assumptions are questionable? I’m sure this is clear to the author, but there are a lot of concepts and assumptions for the reader to keep track of so it would help to spell out the argument a bit more.

  6. Similarly, in the remarks following Eq. (28) it would help to remind the reader of the meaning of the terms therein, and why it is “certainly negligibly small”.

  7. Above Eq.(34), why call this operator an “energy imbalance”? If I understood correctly, there is as much energy spread within each band as there is from one band to the next.

  8. From Fig. 3 on, the figure captions are extremely brief. I don’t know if SciPost has a policy on this, but in other high-quality journals it is required to put enough information in the caption so that a reader can look at the figure and caption and understand what is being plotted. Here that information is buried in the text, often across several paragraphs. I suggest the author to make proper captions and remove the explicit descriptions (of colours and line-styles etc.) from the main text. The main text can just say what is plotted. The caption says how. Quantities plotted should either be given a description or an equation reference in the caption.

  9. In the second-last paragraph of page 22, it would help to remind the reader what q_{t,s}(x_0) is.

  10. This statement in the conclusion: "In fact, by looking at Eq. (18) it even becomes clear that we have derived the stronger decoherent histories condition” should maybe be discussed at that point in the main text.

There are also some minor things that need fixing:

  1. The sentence crossing from page 1 to page 2 is not grammatically well formed.
  2. In the second-last sentence on page 11, “classical” should be “classically”.
  3. On page 16, “fascillitated” should be “facilitated”.
  4. On page 18, “hugh” should be “huge”
  5. First sentence, last paragraph of page 19, “plays” would be better than “played”.

  • validity: high
  • significance: good
  • originality: good
  • clarity: high
  • formatting: reasonable
  • grammar: good

Author:  Philipp Strasberg  on 2023-03-14  [id 3478]

(in reply to Report 1 on 2023-02-28)
Category:
remark
answer to question

I thank the Referee for their positive report and the constructive criticism, which I aim to address in a revision of the manuscript. While I agree to most comments, I like to highlight the following issues:

Referee's point 1.

It is true that some researchers interpret the histories framework as a new interpretation of QM (e.g., Griffiths or Omnes). Others (e.g., Hartle or Gell-Mann) seem in my view less focused on that question, and whether the histories framework really provides a new interpretation has been fiercely debated (see, e.g., the article "Quantum Histories, Mysteries, and Measurements" in Physics Today from 2000). I have emphasized in the manuscript that the question I investigate in Sec. 3 is well posed without any recourse to interpretation issues, and to keep the manuscript focused and uncontroversial I do not plan to discuss interpretations in more detail. It was therefore my intention to speak about the "formalism" or "framework" of consistent/decoherent histories, which I think is the least biased and uncontroversial terminlogy.

Referee's point 5.

A state drawn at random, i.e., completely uncorrelated from the considered observable, is with overwhelming probability an equilibrium state (this is typicality). Nonequilibrium states therefore must be correlated somehow with the considered observable. I will try to make this more precise in the revised manuscript.

Referee's point 6.

Even if all terms scale in the worst case, Eq. (28) definitely contains a left over factor of 1/D. Since D is enormous, it is "certainly negligible small". In general, I here just follow the conventional stat mech terminology to call all quantities scaling exponentially with the particle number (with a negative exponent) "negligible small".

Referee's point 7.

The observable measures whether there is more population in band 1 or band 2, so with respect to the chosen coarse-graining this certainly measures an energy imbalance (at least at weak coupling). It is true that this energy imbalance might be of the order of the imbalance in each band itself. However, the latter can be controlled by \delta E, which I can choose freely. I therefore think the terminology has justification. Since my preference is here to present a transparent mathematical model that is easy to grasp, I do not plan to engage with detailed physical justifications too much.

Anonymous on 2023-03-18  [id 3495]

(in reply to Philipp Strasberg on 2023-03-14 [id 3478])
Category:
remark

Referee here.

Regarding 1, my request was not that you introduce a discussion of interpretations. Rather just to be clear up front "that the question I investigate in Sec. 3 is well posed without any recourse to interpretation issues". That is, to make sure the readers know what to expect from the paper.

Regarding 6, I was not questioning the fact or the terminology, just suggesting that you make it easier for the reader by saying something very much like you do above, "Even if all terms scale in the worst case, Eq. (28) definitely contains a left over factor of 1/D. Since D is enormous -- scaling exponentially with the particle number -- Eq. (28) is certainly negligible small."

---

## Round 1 · Referee Report · Anonymous (Referee 2) · 2023-3-23

Report

This paper develops the formalism of decoherent histories in an interesting and careful manner, emphasizing the emergence of probabilities that satisfy the Kolmogorov consistency conditions as the key to classical behavior, with "classical" defined as meaning that the effects of quantum interference are negligibly small. Numerics with a simple random-matrix model serve to demonstrate the abstract results.

---

## Round 2 · Referee Report · Anonymous (Referee 1) · 2023-4-5

Report

The author has addressed almost all the changes I suggested. It would have been helpful for the author to set out the changes made so that I didn't have to try to find them in the ms. However, I recommend acceptance.

---

## Round 2 · Referee Report · Anonymous (Referee 2) · 2023-4-7

Report

This paper develops the formalism of decoherent histories in an interesting and careful manner, emphasizing the emergence of probabilities that satisfy the Kolmogorov consistency conditions as the key to classical behavior, with "classical" defined as meaning that the effects of quantum interference are negligibly small. Numerics with a simple random-matrix model serve to demonstrate the abstract results.

---

## Round 2 · List of Changes

I followed all the helpful recommendations of Referee 1 (except point 7), and I further included some minor modifications and some additional references due to private correspondences with colleagues. Since the modifications are minor reformulations, and none impacts the validity of the core scientific statements in the manuscript, I refrain from listing them here in detail.

---

## Editorial Decision

published